# Breaking the deadlock: A study on the pathway and effects of reshaping the sustainable marketing capability of Chinese time-honored brands

Xiangyu Li[1,2]*, Danaikrit Inthurit[2], MaoChun Wu[2,3], Linjun Qiu[1]

**1** Faculty of Management and Economics, Chuxiong Normal University, Chuxiong, Yunnan Province, China, **2** International College, Maejo University,Chiang Mai, Thailand, **3** Boya College, Geely University of China, Chengdu, Sichuan, China

* axgg2046@outlook.com

## Abstract

Numerous Chinese Time-Honored Brands are facing decline due to their limited ability to adapt to rapidly changing environments. This study identifies the core barrier as a lack of focus on developing Sustainable Marketing Capability (SMC) within the firms that own these brands. While prior research has examined the link between general marketing capabilities and corporate performance, limited attention has been paid to SMC and its effects. To address this gap, we investigated both the antecedents and outcomes of SMC. Drawing on survey data from managers and selected employees at 47 heritage enterprises in Southwest China, all of which own China Time-Honored Brands, our findings reveal that ambidextrous marketing—comprising proactive marketing exploration and exploitation—significantly enhances SMC. In turn, SMC positively impacts corporate performance and mediates the relationship between ambidextrous marketing and performance by strengthening firms' responsiveness to market and policy conditions. This study provides actionable strategies for heritage enterprises seeking to build sustainable marketing capabilities and offers practical insights into maintaining competitiveness and achieving long-term success.

## 1. Introduction

The advent of digital and smart technologies has driven the emergence of new markets, products, business models, and consumer lifestyles. Numerous Chinese Time-Honored Brands have struggled to adapt to these environmental shifts, resulting in a gradual loss of market presence and relevance over time [1]. To date, only 1,455 brands have been officially recognized by China's Ministry of Commerce, representing 1,341 enterprises [2]. In contrast, there were 16,000 time-honored brands in 1949 [3]. Among the currently recognized brands, only a small number remain competitive [4].

**Data availability statement:** All relevant data are within the manuscript and its Supporting Information files. We also attempted to upload a ZIP archive containing both the raw data and the processed analysis files; however, the upload was not successful due to technical limitations. As a result, we have provided the complete raw dataset (in both original data.xlsx and original data.sav formats), which is sufficient to reproduce all analytical results presented in the manuscript. If reviewers or readers require access to the full set of processed analysis files (including model outputs and intermediate computations), we would be happy to provide them upon request. Please feel free to contact the corresponding author for access.

**Funding:** The author(s) received no specific funding for this work.

**Competing interests:** The authors have declared that no competing interests exist.

This widespread decline is largely attributed to a lack of adaptability in response to digital transformations—ranging from technological upgrades and R&D to product innovation and changes in operational models [5,6]. While some legacy brands, such as Tong Ren Tang [7], have sustained success, many have closed or faded. To counter this trend, scholars have underscored the importance of innovation [8], particularly in heritage branding [9–11]. Despite considerable research on various forms of innovation [9–12], limited attention has been paid to how marketing capabilities function as a foundation for innovation.

Day [13] emphasized that marketing capabilities are essential for identifying and seizing market opportunities, serving as the bedrock for innovation. Similarly, Vorhies and Morgan [14] argued that strong marketing capabilities not only foster innovation but also create a feedback loop that continually enhances a firm's marketing competence. Therefore, the revitalization of China's Time-Honored Brands is a long-term endeavor that hinges on robust marketing practices and sustained innovation. To maintain competitiveness, these brands must remain proactive and avoid complacency [15].

This study focuses on Sustainable Marketing Capability (SMC) as a core internal driver of continuous innovation and long-term competitive advantage for heritage enterprises. SMC does not operate in isolation; rather, it requires firms to systematically coordinate various elements within the marketing system [16]. The development of SMC is contingent upon effective marketing exploration and exploitation [17,18], and its success must be validated in real-world market environments [15].

Although research on sustainable competitive advantage has gained traction in recent years [19–21], studies focusing specifically on SMC are scarce [22,23]. The antecedents of SMC have yet to be thoroughly explored. This study addresses this gap by examining Ambidextrous Marketing as a precursor to SMC. While prior studies have analyzed the relationship between marketing capabilities and Corporate Performance [24–28], few have offered an integrated view that links the antecedents and outcomes of SMC.

Based on these gaps, this study aims to answer the following research questions:

(1) What are the key components of the full development pathway of sustainable marketing capability in firms?

(2) Does sustainable marketing capability mediate the relationship between ambidextrous marketing and corporate performance?

(3) Do market and policy environments moderate the relationship between sustainable marketing capability and corporate performance?

Contingency Theory [29] posits that Time-Honored Brands must adjust both their internal structures and external strategies in response to environmental changes in order to remain adaptable. The Resource-Based View [30] emphasizes that a firm's competitive advantage stems from the scarcity and inimitability of its internal resources. To achieve long-term advantages, firms must continuously enhance the efficiency with which they utilize these resources in a dynamic environment. Organizational Adaptation

Theory [31,32] highlights the importance of constant organizational adjustment in response to environmental changes. Firms can attain sustainability by realigning their structures and processes to meet the evolving external demands. Ambidextrous Learning Theory [33] further suggests that firms should simultaneously engage in the exploration of new opportunities and the exploitation of existing resources. This strategic balance, termed ambidextrous learning, optimizes current capabilities while uncovering future market potential, ultimately improving corporate performance.

To address the aforementioned research questions, this study integrates these theoretical perspectives and proposes a comprehensive framework: "Corporate Adaptability → Sustainable Capability → Corporate Performance." Focusing on Sustainable Marketing Capability (SMC), we examine how ambidextrous marketing behaviors influence SMC development and, in turn, drive corporate performance in the context of China's Time-Honored Brands. This study offers practical guidance for heritage enterprises aiming to develop sustainable capabilities that are both adaptive and strategically relevant.

This study makes four primary contributions to the field of management.

(1) This study develops and empirically validates a closed-loop model linking ambidextrous marketing, sustainable marketing capability, and corporate performance.

(2) It clarifies the mediating role of sustainable marketing capability and its key dimensions in translating dual marketing activities into performance gains.

(3) It highlights the moderating effects of market and policy environments on the relationship between capability and performance.

(4) It provides empirical insights from Southwest China, a region largely underrepresented in current literature, thereby addressing critical gaps in research on the transformation of traditional enterprises.

All enterprises selected for this study are located in Southwest China, a region characterized by distinctive industrial structures, cultural heritage, organizational forms, and brand legacies. Situated at the intersection of market dynamics, policy support, and shifting consumer preferences, these brands experience a structural tension between preserving traditional stability and embracing modern adaptability. Investigating how they cultivate SMC within this complex context offers not only a solution to pressing practical challenges but also an opportunity to uncover culturally grounded pathways for capability evolution.

The remainder of this paper is structured as follows: Section 2 reviews the literature and proposes the hypotheses. Section 3 describes the research methodology. Section 4 presents the results. Section 5 discusses the findings, including the key implications and limitations. Section 6 concludes the study and answers the research questions.

## 2. Literature review and hypotheses

### 2.1. Ambidextrous marketing

The term *ambidexterity* originates from Latin, meaning the ability to use both the left and right hands equally well. Since March [33] introduced the concept of balancing the exploration of new opportunities with the exploitation of existing knowledge in organizational learning, research on ambidexterity has expanded significantly. Kyriakopoulos and Moorman [34] applied this concept to the marketing domain, examining how firms integrate exploratory and exploitative strategies in product development. Exploration involves entering new market segments, adjusting positioning, and experimenting with novel products, channels, and marketing mixes. In contrast, exploitation focuses on refining and optimizing the existing strategies.

From a strategic perspective, Prange and Schlegelmilch [35] emphasized that Marketing Exploration (PLOR) drives real-time innovation in response to market changes, while Marketing Exploitation (PLOI) improves operational efficiency through incremental advancements. Their research showed that the two strategies are complementary, and when combined with strong market orientation, they significantly enhance new product performance. Vorhies et al. [36] further positioned marketing knowledge improvement as a higher-order dynamic capability: PLOR reflects an organization's ability to

acquire new knowledge and develop new skills, while PLOI leverages existing knowledge to maximize value. Both capabilities are essential for sustainable growth.

Ambidextrous Marketing enables firms to adapt to external changes by combining the acquisition of new insights with the efficient use of established resources, which is an embodiment of adaptive learning. However, Levinthal and March [37] warned that excessive focus on exploration can lead to a "failure trap," in which short-term viability is compromised by an overemphasis on future potential. Conversely, overreliance on exploitation may result in a "success trap," that hinders innovation and long-term adaptability. Therefore, achieving a strategic balance is critical.

Subsequent studies have confirmed the importance of this balance. He and Wong [38] examined listed firms in Malaysia and Singapore and found that a high degree of interaction between exploratory and exploitative innovation is positively correlated with sales growth. In contrast, an imbalance between the two had a negative impact. Similarly, Cao et al. [39] found that both balanced and interactive ambidextrous capabilities positively influenced corporate performance in Chinese firms, although the magnitude varied by environmental context.

Nevertheless, some studies have pointed to possible trade-offs. Ho and Lu [40] found that firms attempting to simultaneously achieve high levels of marketing exploration and marketing exploitation often underperformed. This may result from the strain such duality places on organizational resources and managerial focus, leading to inefficiencies or misallocation of efforts. However, Peng et al. [41] showed that a balanced ambidextrous approach among founding teams has a significant positive effect on performance outcomes.

Over the past two decades, research has shifted from viewing exploration and exploitation as competing priorities towards emphasizing their coordination and integration. Recent scholarship highlights the importance of achieving effective ambidexterity through the balanced deployment or synergistic interaction of both activities [42–46].

In summary, this study defines Ambidextrous Marketing as a firm's capability to simultaneously conduct marketing exploration and exploitation in dynamic environments, with the goal of integrating short-term performance improvements and long-term capability development. Specifically, *marketing exploration* refers to the continuous development of innovative strategies that break from existing routines and relying on new market knowledge, experimental thinking, and disruptive practices. In contrast, *marketing exploitation* focuses on enhancing operational efficiency by refining existing processes, optimizing resource use, and building cumulative marketing knowledge. By combining both, firms can strengthen and sustain their brand competitiveness in fast-changing markets.

## 2.2. Sustainable marketing capability

Sustainable Marketing Capability (SMC) builds upon the broader concept of marketing capability, which has long been a focus of strategic marketing research. Day [13], from a resource-based perspective, categorized marketing capabilities into external, internal, and integrated forms, each enabling firms to gain a competitive advantage. Vorhies and Morgan [14] defined marketing capability as an organization's ability to identify, create, and deliver value propositions to target customers to pursue strategic goals.

Although definitions vary, four major theoretical perspectives frame the current understanding of marketing capability:

(1) Resource-Based View: Positions marketing capability as a strategic resource that supports both market management and innovation efforts [13,30,47].

(2) Strategic Perspective: Emphasizes executional ability at the strategic level, linking marketing capability to firm-level competitiveness [22,48,49].

(3) Knowledge-Based View: Focuses on how differentiated knowledge resources drive firm performance [13,14].

(4) Dynamic Capability View: Marketing capability is viewed as a flexible, evolving construct shaped by volatile market environments [47,50,51].

Together, these perspectives expand the theoretical scope and practical applications of marketing capability. From a strategic standpoint, Sustainable Marketing Capability represents a firm's ability to manage the full value exchange process over time. Kotler [22] warned against "marketing myopia" (a short-term mindset) and introduced the concept of sustainable marketing, advocating for a long-term, stakeholder-driven business philosophy that balances the interests of customers, employees, and society.

Kotler also observed that SMC is a defining feature of successful Asian enterprises, and described it through the Sustainable Marketing Enterprise model. Although the model does not explicitly define SMC, subsequent studies have supported its theoretical foundation. While relatively few studies directly use the term "Sustainable Marketing Capability," scholars have explored the sustainability of marketing functions from strategic, operational, and ethical viewpoints.

For example, Hooley [52], from the resource-based view lens, argued that unique and culturally embedded capabilities can generate enduring competitive advantages. Wang and Xu [23] proposed that sustainable marketing capability is built on internal customer satisfaction and externally oriented towards market and competitor needs. Aligning internal processes with external communication channels enhances long-term firm growth.

Wang [53] further noted that marketing culture and financial capital are critical enablers of sustainable marketing in dynamic environments. Kamboj and Rahman [54] demonstrated that marketing capability promotes sustained innovation, especially in market-driven firms. Hunt [55] emphasized the central role of marketing in sustainable economic development, advocating for its integration into corporate values and long-term objectives.

Other studies have added further depth. Sinčić et al. [56] analyzed sustainable marketing orientation through the lenses of strategic integration, social engagement, and ethical responsibility. Appiah-Nimo and Chovancová [57] highlighted the importance of market orientation in sustaining firm performance. Ling et al. [58] linked digital transformation with enhanced innovation and capability development. Scholars have also examined sustainable marketing capability from a triple-bottom-line perspective: economic, social, and environmental sustainability [20,59,60].

This study identifies three defining characteristics of Sustainable Marketing Capability:

(1) Valuable, Unique, and Inimitable: sustainable marketing capability constitutes a core strategic asset that competitors find difficult to replicate.

(2) Dynamic: It evolves through continuous learning, resource integration, and capability accumulation.

(3) Integrated: It reflects the strategic coordination of firms, employees, stakeholders, and society in pursuit of sustainable development goals.

Accordingly, we define Sustainable Marketing Capability as a unique, strategic resource formed through the long-term integration of internal and external assets. It evolves via continuous learning and involves the joint efforts of customers, employees, firms, and stakeholders to drive innovation, respond to market change, and ensure long-term performance. SMC forms a critical foundation for achieving sustainable competitive advantage.

Adopting an internal organizational perspective, this study examines the development of SMC among China's Time-Honored Brands. Drawing on the resource-based view and prior literature [23,24], SMC is conceptualized across three key dimensions:

• Marketing Culture: Shared values, norms, and behaviors shaped by customer and market orientation [61];

• Marketing Learning: The firm's ability to gather, disseminate, and apply knowledge to adapt to market dynamics [13,62];

• Marketing Operation: The executional aspect of strategy, including resource allocation, channel management, customer service, and brand maintenance [14,47].

These capabilities are cultivated through long-term accumulation and function as coherent systems. Collectively, they serve as core marketing assets that are difficult to imitate and essential for sustaining firm competitiveness in dynamic environments.

### 2.3. Ambidextrous marketing and sustainable marketing capability (SMC)

Although few studies have directly examined the relationship between Ambidextrous Marketing and SMC, a review of the existing literature suggests a meaningful theoretical connection. Both constructs have been independently shown to influence Corporate Performance.

For instance, Zhang and Qiu [63] found that exploratory innovation promotes differentiation, whereas exploitative innovation enhances cost efficiency, both of which contribute to improved performance. Kunieda and Takashima [26] argued that adjusting the allocation of resources between exploration and exploitation over time strengthens financial outcomes and market valuation, as reflected in Tobin's Q. Jiang [64] reported that the relationship between marketing ambidexterity and sales growth follows an upward, concave trajectory.

Li and Zheng [65] revealed that exploitative innovation and marketing capabilities significantly enhance performance, while exploratory innovation influences performance indirectly through marketing capabilities. Similarly, Xu and Wang [24] found that the Marketing Culture component of SMC positively affects long-term performance, while Marketing Learning and Marketing Operation influence both short- and long-term outcomes. Zhang et al. [19] further observed that balanced ambidextrous innovation exerts an inverted U-shaped effect on short-term financial performance, yet contributes positively to long-term competitiveness. Ćorić et al. [56] also demonstrated that sustainable marketing orientation boosts the performance of startups.

Taken together, these studies—though focused on various aspects of marketing—imply a potential link between ambidextrous marketing and SMC. However, the specific mechanisms connecting these two constructs have yet to be clearly articulated, indicating a significant gap in the literature. This study addresses this gap by examining how ambidextrous marketing behaviors contribute to the formation and enhancement of SMC within firms.

Beyond the marketing domain, extensive research across disciplines has affirmed a positive relationship between ambidexterity and sustainability. For instance, Lyu et al. [66] confirmed that ambidextrous leadership promotes sustainability outcomes. Zhang et al. [21] found that ambidextrous innovation strategies, when supported by big data capabilities, significantly improve sustainable competitive advantage. Li et al. [67] highlighted the role of ambidextrous orientation in advancing sustainable procurement, while Qian and Peng [6] identified ambidextrous innovation synergy as a mediating factor between dynamic capabilities and sustainable competitive advantage. Sijabat et al. [68] also linked ambidextrous innovation with enhanced performance and competitiveness via entrepreneurial creativity. Zhang et al. [19] provided further theoretical support, emphasizing the critical role of ambidextrous capabilities in driving sustainable innovation.

Although these studies do not directly address sustainable marketing, they collectively lay a robust theoretical foundation for exploring how ambidextrous strategies may influence SMC. Based on this foundation, the following hypotheses are proposed:

**Ha: Ambidextrous Marketing positively influences Sustainable Marketing Capability.**

Ha1: Marketing Exploration has a positive effect on Marketing Culture.

Ha2: Marketing Exploration has a positive effect on Marketing Learning.

Ha3: Marketing Exploration has a positive effect on Marketing Operation.

Ha4: Marketing Exploitation has a positive effect on Marketing Culture.

Ha5: Marketing Exploitation has a positive effect on Marketing Learning.

Ha6: Marketing Exploitation has a positive effect on Marketing Operation.

## 2.4. Sustainable marketing capability and corporate performance

Corporate performance refers to the systematic evaluation of how efficiently and effectively a firm achieves its strategic objectives through resource allocation and operational execution over a defined period. Traditionally, performance has been assessed using financial indicators, such as profitability, cost control, and return on investment [69].

However, with the evolution of research and managerial practices, performance assessment has expanded beyond financial metrics to include multidimensional frameworks incorporating non-financial indicators [70]. This broader approach allows for a more comprehensive and balanced evaluation of firm success. In this study, both financial and non-financial performance measures are considered. Short-term indicators include net profit, sales margin, and cash flow, whereas long-term indicators include sales growth, market share, and new product development. This integrated model enables a holistic assessment of China's Time-Honored Brands within a sustainability-oriented context.

Among the few studies explicitly linking Sustainable Marketing Capability to Corporate Performance, Xu and Wang [24] provide valuable insights. They argue that a strong Marketing Culture enhances satisfaction among internal and external stakeholders, thereby boosting overall performance. Marketing Learning facilitates knowledge sharing and cross-functional collaboration, enabling firms to create greater customer value and secure competitive advantages. Within Marketing Operation, strategic capabilities help firms "win a place in customer memory," while tactical capabilities allow them to "capture market share." These targeted actions directly contribute to improved market positioning and better business outcomes.

Other scholars have provided complementary evidence. For example, Mumel et al. [71] demonstrated that market orientation significantly enhances customer loyalty and improves firm performance, emphasizing its role in sustaining competitiveness. Sampaio et al. [72] highlighted the importance of balancing exploration and exploitation in shaping a firm's long-term financial results and market valuation. Kunieda and Takashima [26] further showed that service quality mediates the link between market orientation and business performance, underscoring the operational pathways through which capabilities influence outcomes.

Based on these insights, this study proposes the following hypotheses:

**Hb: Sustainable Marketing Capability positively influences Corporate Performance.**

Hb1: Marketing Culture has a positive effect on Corporate Performance

Hb2: Marketing Learning has a positive effect on Corporate Performance

Hb3: Marketing Operation has a positive effect on Corporate Performance

## 2.5. The mediating role of sustainable marketing capability

A growing body of research has explored the mediating role of Sustainable Marketing Capability in the relationship between Ambidextrous Marketing and Corporate Performance.

For example, Asree [73] investigated innovation as a mediator between ambidextrous supply chain strategies and performance, showing that innovation capabilities transmit the effect of ambidextrous inputs on performance outcomes. He et al. [45] further distinguished between exploratory and exploitative marketing innovation, finding that Marketing Exploration enhances innovation performance through exploratory, market-oriented innovation, while Marketing Exploitation contributes via exploitative innovation. Their findings emphasize the mediating role of marketing capabilities in shaping innovation-driven performance gains.

Similarly, Zhang et al. [21] demonstrated that ambidextrous innovation strategies partially mediate the relationship between big data capabilities and sustainable competitive advantage. Lyu et al. [66] examined sequential mediation involving employee green creativity and green product innovation, revealing that ambidextrous leadership fosters green

creativity, which in turn enhances sustainable development outcomes. Their results highlight the crucial mediating role of sustainable marketing behavior in linking ambidextrous inputs to sustainability-oriented performance.

Collectively, these studies underscore how ambidextrous innovation—when filtered through marketing or sustainability capabilities—can amplify the performance outcomes. Building on this foundation, the current study proposes that the sustainable marketing capability serves as a mediating mechanism linking ambidextrous marketing strategies to corporate performance.

Based on this reasoning, the following hypotheses are proposed.

**Hc: Sustainable Marketing Capability mediates the relationship between Ambidextrous Marketing and Corporate Performance.**

Hc1: Marketing Culture mediates the effect of Marketing Exploration on Corporate Performance.

Hc2: Marketing Learning mediates the effect of Marketing Exploration on Corporate Performance.

Hc3: Marketing Operation mediates the effect of Marketing Exploration on Corporate Performance.

Hc4: Marketing Culture mediates the effect of Marketing Exploitation on Corporate Performance.

Hc5: Marketing Learning mediates the effect of Marketing Exploitation on Corporate Performance.

Hc6: Marketing Operation mediates the effect of Marketing Exploitation on Corporate Performance.

### 2.6. The moderating role of market and policy environment

Market Environment (ME) refers to the broader set of external conditions that influence a firm's marketing activities. These include shifts in consumer demand, competitor strategies, supplier dynamics, market trends, and economic fluctuations.

In contrast, the Policy Environment (PE) encompasses the institutional and regulatory frameworks that shape business behavior. These include legal systems, administrative mechanisms, and public policy tools, such as economic, industrial, and environmental policies, that directly or indirectly affect corporate strategy and performance [74]. According to Institutional Theory [75,76], firm decision-making is shaped by the external environment, which in turn impacts Corporate Performance. Among these external factors, both ME and PE are critical forces that interact with internal capabilities of the firm. This section explores how ME and PE moderate the relationship between Sustainable Marketing Capability (SMC) and Corporate Performance.

Prior research has provided empirical support for these moderating effects. Peng et al. [77] found that market conditions significantly influence the relationship between marketing innovation and performance. In particular, technological turbulence positively moderated the impact of market-driven marketing innovation on performance, underscoring the dynamic role of the external market.

Similarly, Appiah-Nimo and Chovancová [57] examined how internal and external business environments, particularly market orientation, interact with marketing strategies to influence sustainable performance. Their findings confirm that alignment with market conditions enhances the effectiveness of sustainable marketing initiatives.

Regarding the policy environment, Luan et al. [78] demonstrated that supportive environmental policies strengthen the relationship between firm strategies and economic sustainability. Tshuma [79] emphasized that green marketing strategies are often shaped by policy pressures and shifting consumer expectations, reinforcing the moderating role of PE in marketing performance linkages.

Taken together, these studies suggest that favorable market and policy environments can amplify the SMC's positive effects on corporate performance. Therefore, companies must consider both ME and PE when designing and implementing sustainable marketing strategies. Based on these insights, we propose the following hypothesis:

**Hd: A favorable Market Environment strengthens the effect of Sustainable Marketing Capability on Corporate Performance.**

Hd1: Market Environment strengthens the effect of Marketing Culture on Corporate Performance.

Hd2: Market Environment strengthens the effect of Marketing Learning on Corporate Performance.

Hd3: Market Environment strengthens the effect of Marketing Operation on Corporate Performance.

**He: A favorable Policy Environment strengthens the effect of Sustainable Marketing Capability on Corporate Performance.**

He1: Policy Environment strengthens the effect of Marketing Culture on Corporate Performance.

He2: Policy Environment strengthens the effect of Marketing Learning on Corporate Performance.

He3: Policy Environment strengthens the effect of Marketing Operation on Corporate Performance.

At this stage, all the research hypotheses were fully developed. Fig 1 presents the conceptual research framework, including all hypothesized paths.

## 3. Methodology

### 3.1. Research design

This study adopted a quantitative research approach based on survey data to empirically examine the proposed model. Structural Equation Modeling was used to test the causal relationships among ambidextrous marketing, sustainable marketing capability, and corporate performance (Research Question 1). Bootstrapping was employed to assess the mediating role of sustainable marketing capability (Research Question 2). To explore the contextual effects, Latent Moderated Structural Equations were applied to evaluate how market and policy environments moderate these relationships

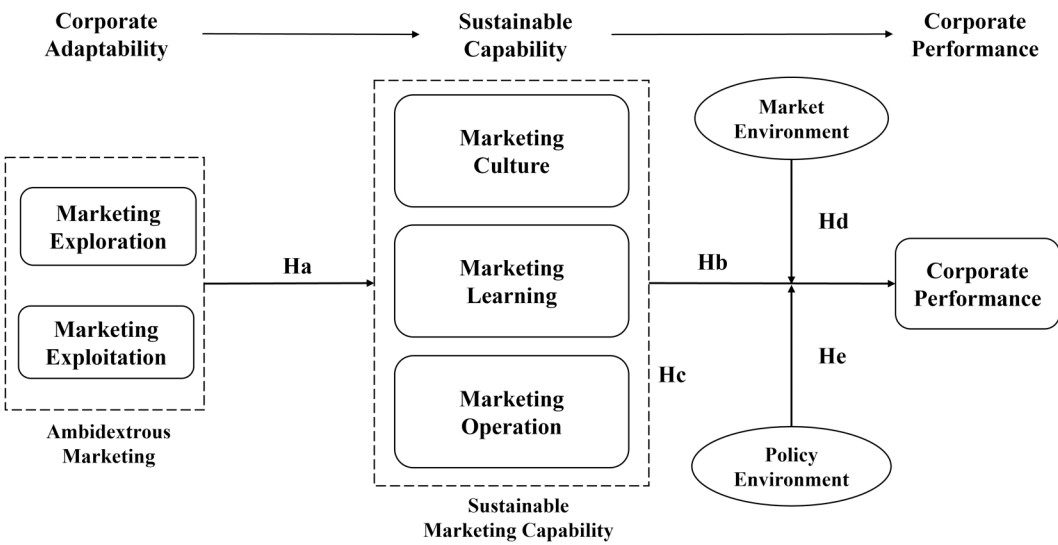

**Fig 1. The research model.**

(Research Question 3). This multi-method design enables a comprehensive and rigorous analysis of both direct and indirect effects and environmental contingencies.

## 3.2. Ethics statement

This study was approved by the Ethics Committee of Chuxiong Normal University. All participants were fully informed of the study's purpose, procedures, and their right to participate voluntarily. Informed verbal consent was obtained prior to the completion of the questionnaire. Data were collected via online platforms, email, and face-to-face distribution. No minors were involved in this study, and all procedures complied with institutional and ethical research guidelines.

## 3.3. Measurement

To ensure the reliability and validity of the results, this study employed well-established measurement scales covering eight variables: six core constructs and two moderating variables. All items were adapted from validated scales published in peer-reviewed international journals. A rigorous translation and back-translation process was conducted, followed by an expert review by scholars in marketing, strategic management, financial management, and industrial economics. Confirmatory factor analysis (CFA) and reliability testing were performed to validate the accuracy and consistency of the scales. All items were measured using a 7-point Likert scale ranging from 1 ("strongly disagree") to 7 ("strongly agree"). The detailed items and their corresponding dimensions are listed in the "Variables and Items" column of Table 3.

(1) Ambidextrous Marketing: This variable consists of two dimensions: Marketing Exploration and Marketing Exploitation. The scale includes 10 items adapted from Vorhies [36], Peng et al. [41], and Wang and Sun [80], which assess the extent of dual marketing strategies in firms.

(2) Sustainable Marketing Capability: This variable is categorized into three components: Marketing Culture, Marketing Learning, and Marketing Operation. Marketing Culture was adapted from Narver and Slater [81], Conduit and Mavondo [82], and Wei [16], with five items. Marketing Learning, derived from the work of Narver & Slater [81], and Hult & Ferrell [83], also comprises five items. Marketing Operation were developed based on the research of Dess and Beard [84], Doty et al. [85], and Vorhies et al. [86], and included five items.

(3) Corporate Performance: This variable is measured subjectively. Respondents were asked to rate their firm's performance over the past three years, compared to key competitors. The 10-item scale was derived from Govindarajan [87], Dess and Beard [84], Celuch et al. [88], Zehir et al. [89], and Xu and Wang [24], and covers both short-term and long-term performance dimensions. Prior studies have confirmed strong correlations between subjective and objective performance metrics.

(4) Moderating Variables: Policy Environment was measured using a 5-item scale adapted from Li and Atuahene-Gima [90] and Li et al. [91]. The Market Environment was based on the studies of Waldman et al. [92], Lu and Yang [93], and Jiang and Ma [94].

## 3.4. Sampling

This study focuses on China Time-Honored Brand enterprises located in Southwest China, specifically in Sichuan Province, Chongqing Municipality, Yunnan Province, Guizhou Province, and the Tibet Autonomous Region. According to the Ministry of Commerce's official directory, there are 138 recognized brands in this region, represented by 128 enterprises across 26 industries (see Table 1).

**Table 1. Distribution of time-honored brands, enterprises, and industries in Southwest China.**

| Province | Number of Brands | Number of Enterprises | Number of Industries |
|----------|------------------|-----------------------|----------------------|
| Sichuan | 57 | 48 | 16 |
| Chongqing | 31 | 30 | 17 |
| Yunnan | 31 | 31 | 12 |
| Guizhou | 17 | 17 | 6 |
| Tibet | 2 | 2 | 1 |
| Total | 138 | 128 | 26 |

Three key reasons justify the selection of this region.

(1) Industrial representativeness: Time-Honored Brands in Southwest China are concentrated in traditional sectors such as food manufacturing, pharmaceuticals, biotechnology, liquor production, consumer services, and specialty tea. These sectors reflect national industrial patterns, offering a representative cross-section of legacy enterprises.

(2) Environmental relevance: Similar to their counterparts elsewhere, these firms face mounting challenges in developing Sustainable Marketing Capability amid increasing environmental volatility, making them highly pertinent to this study's research focus.

(3) Contextual uniqueness: The region's distinct cultural and economic contexts have shaped unique brand characteristics. However, empirical studies on time-honored enterprises in this area remain limited, underscoring the need to address this significant research gap.

To ensure scientific validity and representative coverage, we formulated a planned sample of 96 enterprises using Cochran's formula for sample size estimation [95]. The sample was proportionally allocated based on the number of time-honored brands in each province as follows.

• Sichuan Province: 36 enterprises

• Chongqing City: 22 enterprises

• Yunnan Province: 23 enterprises

• Guizhou Province: 17 enterprises

• Tibet Autonomous Region: 2 enterprises

This sampling plan provided a balanced foundation for data collection across the five provinces and supported a comprehensive investigation of the regional brand revitalization efforts.

## 4. Results

### 4.1. Sample characteristics

The survey was conducted from January to April 2024 among the targeted China Time-Honored Brand enterprises. A total of 1,200 questionnaires were distributed through telephone outreach, email, and online platforms such as Wenjuanxing. 542 responses were received, of which 352 were deemed valid after screening for completeness and consistency. These valid responses were collected from 47 enterprises that agreed to participate. Although the original sampling plan targeted 96 firms, actual participation was limited due to constraints related to access, availability and willingness. This yields a participation rate of 48.96% and an overall effective response rate of 29.33% (352/1200), providing a robust empirical

foundation for subsequent analysis. Descriptive statistics for the valid responses are summarized in Table 2 and detailed below.

(1) Demographics: Among the respondents, 196 were male (55.7%) and 156 were female (44.3%). Most respondents were affiliated with private enterprises (289 respondents, 82.1%), while 63 respondents (17.9%) were affiliated with state-owned enterprises. Regarding organizational roles, 195 were junior managers (55.4%), 123 were middle managers (34.9%), 21 were senior managers (6.0%), and 13 were general staff (3.7%). Notably, 96.3% of the respondents held managerial positions, aligning well with the study's target population.

(2) Enterprise Size: Respondents were distributed across medium-sized enterprises (163 individuals, 46.3%), large enterprises (125, 35.5%), small enterprises (37, 10.5%), and micro-enterprises (27, 7.7%).

(3) Regional Distribution: Geographically, 130 respondents (36.9%) were from Sichuan Province, 95 (27.0%) from Yunnan Province, 85 (24.1%) from Chongqing, 32 (9.1%) from Guizhou, and 10 (2.8%) from the Tibet Autonomous Region.

(4) Industry Distribution: The sample covered a diverse set of industries, with the largest group in food manufacturing (146 individuals, 41.5%), followed by pharmaceuticals & biotechnology (57, 16.2%), consumer services (55, 15.6%), liquor (31, 8.8%), and refined tea (18, 5.1%). Respondents from other industries numbered fewer than 10 per sector.

In summary, the valid sample offers comprehensive coverage across gender, firm size, management level, geographic location and industry sector. This diversity ensures that the dataset accurately reflects the structural and contextual characteristics of China's time-honored enterprises, providing a strong empirical foundation for subsequent analyses.

Table 2. Results of descriptive analysis (N = 352).

| Category | Frequency | % | Category | Frequency | % |
|---|---|---|---|---|---|
| **Participant Gender** | | | **Firm Type** | | |
| Male | 196 | 55.7 | Private | 289 | 82.1 |
| Female | 156 | 44.3 | State-Owned | 63 | 17.9 |
| **Respondent's Position** | | | **Firm Size** | | |
| Frontline Management | 195 | 55.4 | Medium enterprise | 163 | 46.3 |
| Middle Management | 123 | 34.9 | Large enterprise | 125 | 35.5 |
| Senior Management | 21 | 6 | Small enterprise | 37 | 10.5 |
| Regular Employees | 13 | 3.7 | Microenterprise | 27 | 7.7 |
| **Firm Location** | | | **Firm Industry** | | |
| Sichuan | 130 | 36.9 | Food Manufacturing | 146 | 41.5 |
| Yunnan | 95 | 27 | Pharmaceuticals & Biotechnology | 57 | 16.2 |
| Chongqing | 85 | 24.1 | Consumer Services | 55 | 15.6 |
| Guizhou | 32 | 9.1 | Liquor | 31 | 8.8 |
| Tibet | 10 | 2.8 | Refined Tea | 18 | 5.1 |
| **Firm Industry** | | | **Firm Industry** | | |
| Retailing | 9 | 2.6 | Catering | 3 | 0.9 |
| Agricultural and Sideline Food Processing | 7 | 2 | Diversified Financial Services | 2 | 0.6 |
| Mining | 5 | 1.4 | Health and Social Work | 2 | 0.6 |
| Trade and Commerce Services | 5 | 1.4 | Raw Materials | 2 | 0.6 |
| Culture, Sports, and Entertainment | 4 | 1.1 | Agriculture, Forestry, Animal Husbandry, and Fish | 1 | 0.3 |
| Real Estate Services | 4 | 1.1 | Chemicals | 1 | 0.3 |

## 4.2. Reliability and validity

(1) Reliability Analysis: This study employed Cronbach's α coefficient to evaluate the internal consistency of the measurement scales. Eight primary constructs were tested: Marketing Exploration, Marketing Exploitation, Marketing Culture, Marketing Learning, Marketing Operation, Corporate Performance, Market Environment, and Policy Environment. The Cronbach's α values were 0.861, 0.844, 0.877, 0.860, 0.862, 0.944, 0.875, and 0.879, respectively. All values exceeded the commonly accepted threshold of 0.70, indicating strong internal reliability across all constructs [96].

(2) Validity Analysis: Validity was assessed using content validity and construct validity procedures.

To ensure content validity, all measurement items were adapted from internationally recognized scales that have been validated in Chinese empirical contexts. A pilot study was conducted on a subset of respondents to ensure contextual appropriateness. Expert feedback from scholars in marketing, strategy, and industrial management was incorporated to refine the item wording and structure.

For construct validity, Confirmatory Factor Analysis was conducted using AMOS (see Table 3). The standardized factor loadings (SFL) ranged from 0.686 to 0.851, all exceeding the recommended threshold of 0.50 [97], indicating satisfactory item convergence [97].

Furthermore, the Average Variance Extracted (AVE) values for each construct exceeded 0.50, and the Composite Reliability (CR) values were all above 0.70. These results confirm both convergent validity and strong internal construct reliability [98].

In addition, the results of the discriminant validity test (see Table 4) showed that the square root of each construct's AVE was greater than its correlations with other constructs. This demonstrates that each construct is distinct and that discriminant validity has been adequately established [99].

## 4.3. Means, standard deviations, and correlation analysis of variables

Table 4 reports the means, standard deviations, and Pearson correlation coefficients for all the key variables. The results show that the average values across constructs were relatively high, suggesting that Time-Honored Brand enterprises generally exhibit strong performance in the measured domains. However, the moderate variation observed—particularly in the ambidextrous marketing dimensions—indicates room for improvement in the adoption and execution of these strategies.

The data imply that strengthening ambidextrous marketing practices may enhance firms' Sustainable Marketing Capability, which is likely to further improve Corporate Performance. Correlation analysis revealed significant positive associations among all major variables, consistent with the conceptual framework of this study. These findings offer initial empirical support for the hypothesized relationships and provide a solid statistical foundation for subsequent hypothesis testing and structural model validation.

## 4.4. Structural model analysis

The structural model analysis confirmed that all constructs met acceptable thresholds for reliability and validity. AMOS was used to perform path analysis to test hypotheses Ha and Hb, with the standardized path coefficients and model fit indices presented in Table 5.

Specifically:

(1) Marketing Exploration (PLOR) had a significant positive impact on Marketing Culture (MC) ($\beta = 0.426$, $p < 0.001$), Marketing Learning (ML) ($\beta = 0.352$, $p < 0.001$), and Marketing Operation (MO) ($\beta = 0.359$, $p < 0.001$). Thus, hypotheses Ha1–Ha3 are supported.

**Table 3. Measurement items and result of reliability and validity.**

| Variables and Items | SFL | AVE | CR | Cronbach's α |
|---|---|---|---|---|
| **1. Marketing Exploration (PLOR)** | | 0.554 | 0.861 | 0.861 |
| We introduce bold, adventurous, or avant-garde marketing processes | 0.804 | | | |
| We will consistently develop innovative marketing processes that starkly differ from past marketing approaches. | 0.730 | | | |
| We will continuously apply market knowledge to devise entirely new strategies that diverge from the existing marketing processes | 0.703 | | | |
| We employ market knowledge to break traditional patterns and create novel marketing processes that have not been utilized before | 0.734 | | | |
| We will continually acquire new marketing knowledge or skills that are groundbreaking for the company and the entire industry | 0.748 | | | |
| **2. Marketing Exploitation (PLOI)** | | 0.520 | 0.843 | 0.844 |
| We will focus on revolutionizing marketing processes to enhance efficiency and imple-mentation effectiveness | 0.687 | | | |
| We will continually examine the information from existing projects and learning experi-ences to improve our established marketing processes | 0.717 | | | |
| Throughout the development of new marketing processes, we consistently adhere to and adapt to existing concepts | 0.741 | | | |
| We will progressively refine or elevate the existing marketing processes | 0.703 | | | |
| We excel in summarizing and distilling current marketing experiences and accumulating systematic marketing knowledge | 0.757 | | | |
| **3. Marketing Culture (MC)** | | 0.581 | 0.873 | 0.877 |
| The company's competitive advantage is built upon a thorough understanding of customer needs | 0.851 | | | |
| Employees who provide excellent service to customers can receive corresponding rewards within the company | 0.686 | | | |
| A company can swiftly respond to competitive actions that pose threats | 0.761 | | | |
| Each department of the company is capable of providing products and services with genu-ine value to its associated departments | 0.756 | | | |
| During cross-departmental collaborations, departments treat each other as customers | 0.747 | | | |
| **4. Marketing Learning (ML)** | | 0.552 | 0.860 | 0.860 |
| The company can quickly identify changes in customer product preferences | 0.751 | | | |
| The company conducts consumer evaluations of its products and services at least once a year | 0.736 | | | |
| When recognizing customer expectations for product or service improvements, all rele-vant departments in the company collaborate to meet these needs | 0.731 | | | |
| Supervisors from each department of the company regularly visit customers or potential customers | 0.733 | | | |
| Managers in the company know how to motivate each employee to create value for the customers | 0.762 | | | |
| **5. Marketing Operations (MO)** | | 0.558 | 0.863 | 0.862 |
| The company's management clearly articulated a strategic approach to achieving the mar-keting objectives | 0.746 | | | |
| The company's marketing strategy aligns with the current market conditions | 0.743 | | | |
| The company gains a competitive advantage in the market by offering differentiated prod-ucts or services | 0.687 | | | |
| The company's marketing mix strategy is more effective than that of its competitors | 0.791 | | | |
| The company established long-term relationships with customers through the sale of products and services | 0.763 | | | |

*(Continued)*

**Table 3.** (Continued)

| Variables and Items | SFL | AVE | CR | Cronbach's α |
|---|---|---|---|---|
| **6. Corporate performance (PERF)** | | 0.629 | 0.944 | 0.944 |
| Net Profit | 0.813 | | | |
| Sales Profit Margin | 0.780 | | | |
| Cash Flow | 0.774 | | | |
| Return on Investment (ROI) | 0.804 | | | |
| Operating Costs | 0.807 | | | |
| Sales Growth Rate | 0.808 | | | |
| Market Share | 0.776 | | | |
| Development of New Products | 0.779 | | | |
| Market Expansion | 0.780 | | | |
| Research and Development Achievements | 0.807 | | | |
| **7. Market Environment (ME)** | | 0.576 | 0.869 | 0.875 |
| Customer demand has rapidly changed | 0.822 | | | |
| It is difficult to predict market competition | 0.743 | | | |
| Competition among peers has intensified | 0.742 | | | |
| Most new products in the market have been developed through technological breakthroughs | 0.731 | | | |
| The pace of technological change in this industry is rapid | 0.752 | | | |
| **8. Policy Environment (PE)** | | 0.591 | 0.878 | 0.879 |
| The government provides policies and projects conducive to company development | 0.814 | | | |
| The government provides the necessary technical information and support to our company | 0.786 | | | |
| Government provides direct fiscal policies to our company, including taxation and government subsidies | 0.773 | | | |
| The government encourages companies to protect intellectual property rights | 0.726 | | | |
| The government provides the necessary legal support for companies to enter new markets | 0.742 | | | |

**Table 4. Discriminant validity: Pearson correlation and AVE square root.**

| Variables | Mean | SD | PLOR | PLOI | MC | ML | MO | PERF | ME | PE |
|---|---|---|---|---|---|---|---|---|---|---|
| **PLOR** | 5.536 | 0.616 | 0.744 | | | | | | | |
| **PLOI** | 5.554 | 0.596 | 0.460** | 0.721 | | | | | | |
| **MC** | 5.503 | 0.654 | 0.460** | 0.356** | 0.762 | | | | | |
| **ML** | 5.571 | 0.617 | 0.394** | 0.409** | 0.420** | 0.743 | | | | |
| **MO** | 5.540 | 0.598 | 0.437** | 0.479** | 0.396** | 0.429** | 0.747 | | | |
| **PERF** | 5.636 | 0.641 | 0.439** | 0.353** | 0.432** | 0.509** | 0.457** | 0.793 | | |
| **ME** | 5.514 | 0.726 | −0.256** | −0.072 | −0.561** | −0.276** | −0.225** | −0.172** | 0.759 | |
| **PE** | 5.426 | 0.622 | −0.164** | −0.051 | −0.390** | −0.188** | −0.116* | −0.103 | 0.668** | 0.769 |

Notes:

**Significance at the 0.01 level (two-tailed). The diagonal numbers represent the square root of the AVE for each factor.

**Table 5. Results of structural model analysis.**

| Variable Relationships | Estimates | P-Value | Hypothesis | Result |
|---|---|---|---|---|
| PLOR→MC | 0.426 | *** | Ha1 | Accepted |
| PLOR→ML | 0.352 | *** | Ha2 | Accepted |
| PLOR→MO | 0.359 | *** | Ha3 | Accepted |
| PLOI→MC | 0.268 | *** | Ha4 | Accepted |
| PLOI→ML | 0.363 | *** | Ha5 | Accepted |
| PLOI→MO | 0.457 | *** | Ha6 | Accepted |
| MC→PERF | 0.198 | *** | Hb1 | Accepted |
| ML→PERF | 0.374 | *** | Hb2 | Accepted |
| MO→PERF | 0.247 | *** | Hb3 | Accepted |

$\chi^2 = 720.844$; $df = 544$; $\chi^2/df = 1.325$; RMSEA = 0.030; GFI = 0.900; NFI = 0.903; TLI(NNFI) = 0.972; CFI = 0.974; *** $p < 0.001$

(2) Marketing Exploitation (PLOI) also had a significant positive impact on MC ($\beta = 0.268$, $p < 0.001$), ML ($\beta = 0.363$, $p < 0.001$), and MO ($\beta = 0.457$, $p < 0.001$). Thus, hypotheses Ha4–Ha6 are confirmed. These results support the positive role of ambidextrous marketing in enhancing Sustainable Marketing Capability (SMC), thereby verifying Ha.

(3) The results also revealed that MC ($\beta = 0.198$, $p < 0.001$), ML ($\beta = 0.374$, $p < 0.001$), and MO ($\beta = 0.247$, $p < 0.001$) had significant positive impacts on Corporate Performance (PERF). This indicates that the improvement in SMC can effectively promote PERF, thus supporting hypotheses Hb1–Hb3 and confirming hypothesis Hb.

Finally, the model fit indices indicated a good overall fit: $\chi^2/df$, RMSEA, GFI, NFI, NNFI, and CFI all met conventional benchmarks. These results demonstrate that the structural model is statistically sound and substantively valid, thereby supporting the theoretical framework developed in this study.

## 4.5. Mediation analysis

To examine the mediating role of Sustainable Marketing Capability (SMC) in the relationship between Ambidextrous Marketing and Corporate Performance (PERF), this study applied the bootstrap method using AMOS. A bias-corrected confidence interval of 95% was employed, based on 5,000 bootstrap samples. The mediation results are presented in Table 6.

(1) In the path exploring the impact of Marketing Exploration (PLOR) on PERF, Marketing Culture (MC), Marketing Learning (ML), and Marketing Operation (MO) showed significant mediating effects. Specifically, in the path " PLOR → MC → PERF," the bootstrap result showed an effect value of 0.084. The upper and lower bounds of the confidence intervals using both the Bias-corrected and Percentile methods did not contain 0, with a p-value of less than 0.05, indicating a significant mediating effect of MC in this path. In the path " PLOR → ML → PERF," the effect value was 0.132, with similar confidence intervals not containing 0 and a p-value of less than 0.001, confirming the significant mediating effect of ML. For the path " PLOR → MO → PERF," the effect value was 0.089, and the confidence intervals did not include 0, with a p-value of less than 0.001, demonstrating the significant mediating effect of MO.

(2) Similarly, in the path examining the effect of Marketing Exploitation (PLOI) on Corporate Performance (PERF), Marketing Culture (MC), Marketing Learning (ML), and Marketing Operation (MO), significant mediating effects were observed. In the path "PLOI → MC → PERF," the effect value was 0.053, with confidence intervals not containing 0 and a p-value of less than 0.05, indicating that MC had a significant mediating effect. In the path "PLOI → ML → PERF," the effect value was 0.136, the confidence intervals did not include 0, and the p-value was less than 0.001, demonstrating a significant mediating effect of the ML. In the path "PLOI → MO → PERF," the effect value was 0.113,

**Table 6. Mediation analysis results.**

| Path | Estimate | SE | Bias-corrected 95% CI | | | Percentile 95% CI | | | Hypothesis | Result |
|---|---|---|---|---|---|---|---|---|---|---|
| | | | Lower | Upper | P | Lower | Upper | P | | |
| PLOR→MC→PERF | 0.084 | 0.037 | 0.028 | 0.178 | * | 0.022 | 0.169 | * | Hc1 | Accepted |
| PLOR→ML→PERF | 0.132 | 0.041 | 0.064 | 0.228 | *** | 0.058 | 0.218 | *** | Hc2 | Accepted |
| PLOR→MO→PERF | 0.089 | 0.032 | 0.037 | 0.166 | *** | 0.032 | 0.156 | *** | Hc3 | Accepted |
| PLOI→MC→PERF | 0.053 | 0.022 | 0.018 | 0.107 | * | 0.013 | 0.099 | ** | Hc4 | Accepted |
| PLOI→ML→PERF | 0.136 | 0.041 | 0.068 | 0.232 | *** | 0.061 | 0.222 | *** | Hc5 | Accepted |
| PLOI→MO→PERF | 0.113 | 0.038 | 0.052 | 0.206 | *** | 0.045 | 0.196 | *** | Hc6 | Accepted |

* $p < 0.05$; ** $p < 0.01$; *** $p < 0.001$

with confidence intervals excluding 0 and a p-value of less than 0.001, confirming that MO had a significant mediating effect.

In summary, Hc1 through Hc6 are supported, indicating that Sustainable Marketing Capability plays a significant mediating role in the relationship between ambidextrous marketing and Corporate Performance, thus confirming Hc's validity.

## 4.6. Moderating effects test

AMOS was used to examine the moderating effects of the Market Environment (ME) and Policy Environment (PE) on the relationship between Sustainable Marketing Capability (SMC) and Corporate Performance (PERF). The detailed results are presented in Table 7.

(1) In the path "MC * ME → PERF," the interaction coefficient was 0.229 ($p < 0.001$), indicating that ME positively moderated the effect of Marketing Culture (MC) on PERF, as shown in Fig 2. Similarly, in the path "ML * ME → PERF," the interaction coefficient is 0.210 ($p < 0.05$), showing a significant positive moderating effect of ME on the relationship between Marketing Learning (ML) and PERF, as shown in the slope chart in Fig 3. Finally, in the path "MO * ME → PERF," the interaction coefficient is 0.322 ($p < 0.001$), indicating that the positive moderating effect of ME on the relationship between Marketing Operation (MO) and PERF is the most significant in Fig 4. Hence, hypotheses Hd1–Hd3 are supported, validating hypothesis Hd: ME positively moderates the effect of SMC on PERF.

(2) In the path "MC * PE → PERF," the interaction coefficient was 0.186 ($p < 0.05$), indicating that PE positively moderated the effect of MC on PERF. The slope chart is shown in Fig 5. Similarly, in the path "ML * PE → PERF," the interaction coefficient was 0.184 ($p < 0.05$), demonstrating that PE positively moderated the relationship between ML and PERF,

**Table 7. Results of interaction relationship tests.**

| Variable Relationships | Estimates | P-Value | Hypothesis | Result |
|---|---|---|---|---|
| MC*ME→PERF | 0.229 | *** | Hd1 | Accepted |
| ML*ME→PERF | 0.210 | * | Hd2 | Accepted |
| MO*ME→PERF | 0.322 | *** | Hd3 | Accepted |
| MC*PE→PERF | 0.186 | * | He1 | Accepted |
| ML*PE→PERF | 0.184 | * | He2 | Accepted |
| MO*PE→PERF | 0.296 | *** | He3 | Accepted |

* $p < 0.05$; ** $p < 0.01$; *** $p < 0.001$

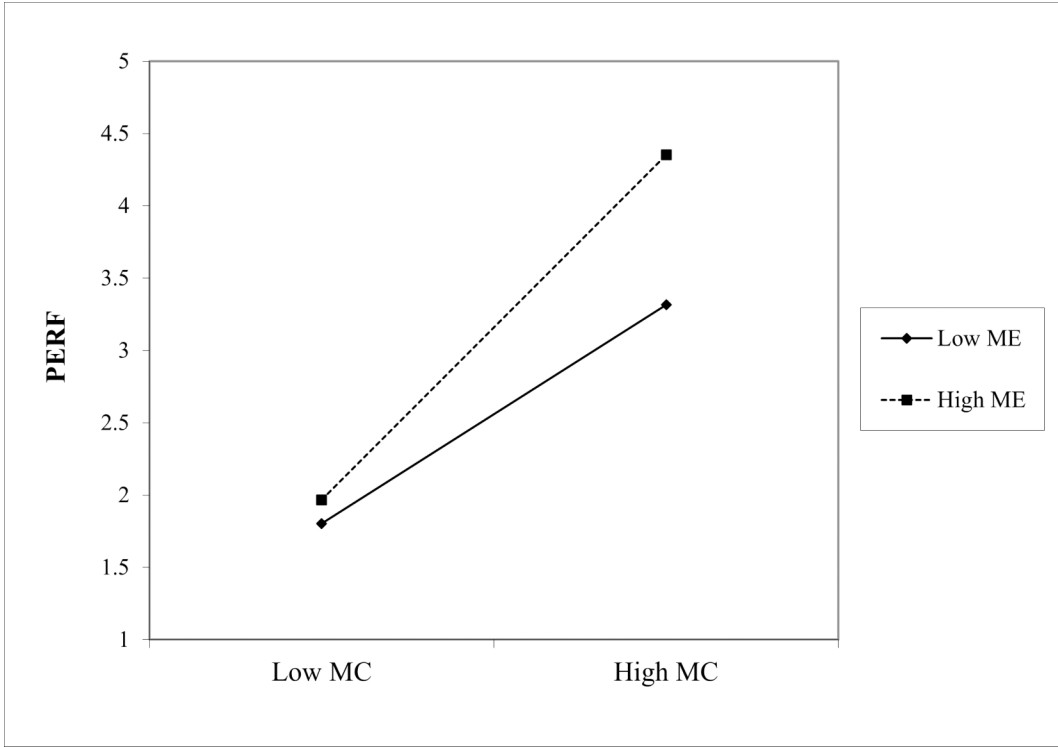

**Fig 2. Moderating effect of ME between MC and PERF.**

as illustrated in Fig 6. Furthermore, in the path "MO * PE → PERF," the interaction coefficient is 0.296 (p<0.001), further confirming that PE positively moderates the relationship between MO and PERF, with the corresponding slope chart shown in Fig 7. Therefore, hypotheses He1–He3 are supported, validating hypothesis He, which states that PE plays a positive moderating role in the relationship between SMC and PERF.

At this point, all the hypotheses proposed in this study have been empirically tested. The full set of findings is summarized in Fig 8, which visually integrates all mediation and moderation pathways. In the figure, the upper-left corner of each SMC construct denotes its mediating role in the path from Marketing Exploration to PERF, while the lower-right corner represents its mediating effect in the path from Marketing Exploitation to PERF.

## 5. Discussion

### 5.1. Sample characteristics and contextual influences

This study focuses on 47 China Time-Honored Brand enterprises located in five provinces of Southwest China: Sichuan, Chongqing, Yunnan, Guizhou, and the Tibet Autonomous Region. These firms are characterized by strong regional cultural embeddedness and deeply rooted brand heritage. Most operate in traditional industries such as baijiu (Chinese liquor), premium tea, traditional Chinese medicine, and ethnic specialty foods. Their brand development follows an integrated trajectory of "region–craftsmanship–culture," aligning with local identity systems. Iconic brands such as Moutai, Duyun Maojian, and Zhangfei Beef exemplify this approach, embodying not only functional consumption but also deep symbolic meaning, demonstrating a sustainable marketing capability that balances heritage with innovation.

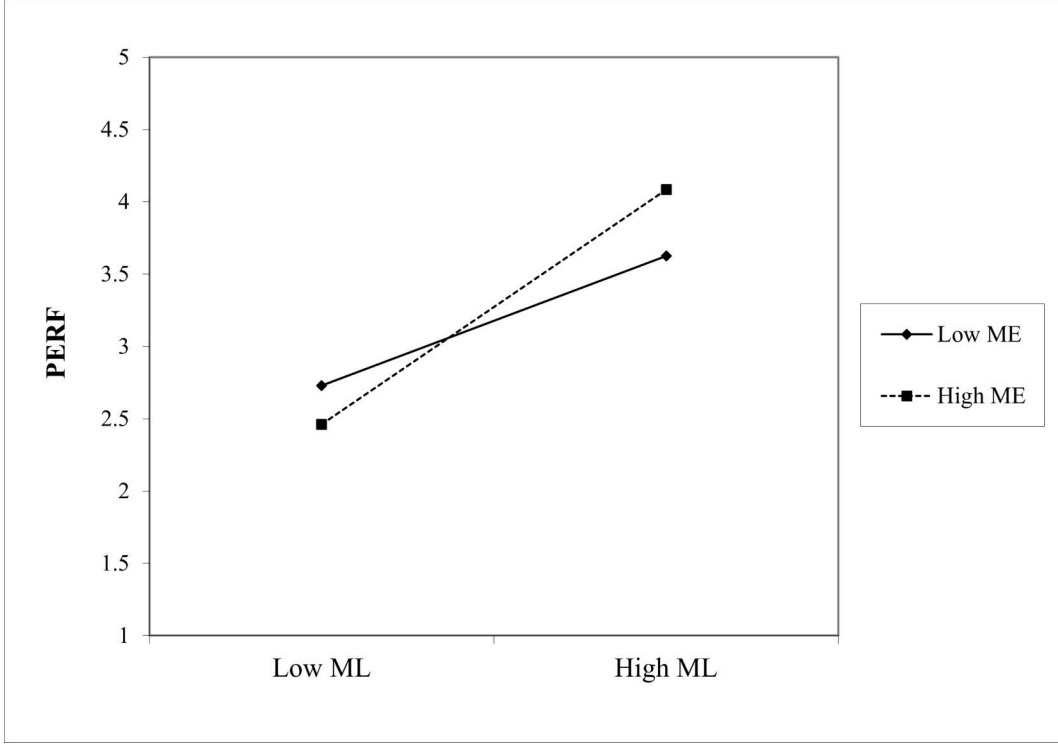

**Fig 3. Moderating effect of ME between ML and PERF.**

Simultaneously, as a relatively less developed economic zone, Southwest China faces challenges in digital transformation, consumption upgrading, and policy implementation. The survey results indicate that the sampled firms tend to show stronger competence in marketing exploitation than in marketing exploration, suggesting a reliance on accumulated experience, operational routines, and standardized processes. However, this configuration also reflects a cautious approach to market expansion and technology adoption, which may limit their responsiveness to dynamic market conditions.

Moreover, local industrial policies significantly influence the performance trajectories of these enterprises. In provinces such as Sichuan and Guizhou, government support for traditional sectors has enhanced firms' marketing operation capabilities but has also increased their dependency on external factors such as regulatory shifts and market volatility, a pattern confirmed through the moderation analysis.

Overall, the interplay between cultural heritage, institutional conditions, and environmental dynamics shapes the pathways through which these brands cultivate Sustainable Marketing Capability. This highlights the importance of considering regional heterogeneity and contextual embeddedness when interpreting findings or attempting broader generalizations.

### 5.2. Research finding

**5.2.1. The impact of ambidextrous marketing on sustainable marketing capability.** Anchored in the theoretical logic of "corporate adaptability → sustainable capability → corporate performance," this study constructs a capability development model for China's Time-Honored Brands by linking Ambidextrous Marketing—comprised of Marketing Exploration (PLOR) and Marketing Exploitation (PLOI)—with Sustainable Marketing Capability (SMC) and firm performance. Empirical results confirm that both PLOR and PLOI exert significant positive effects on all three dimensions

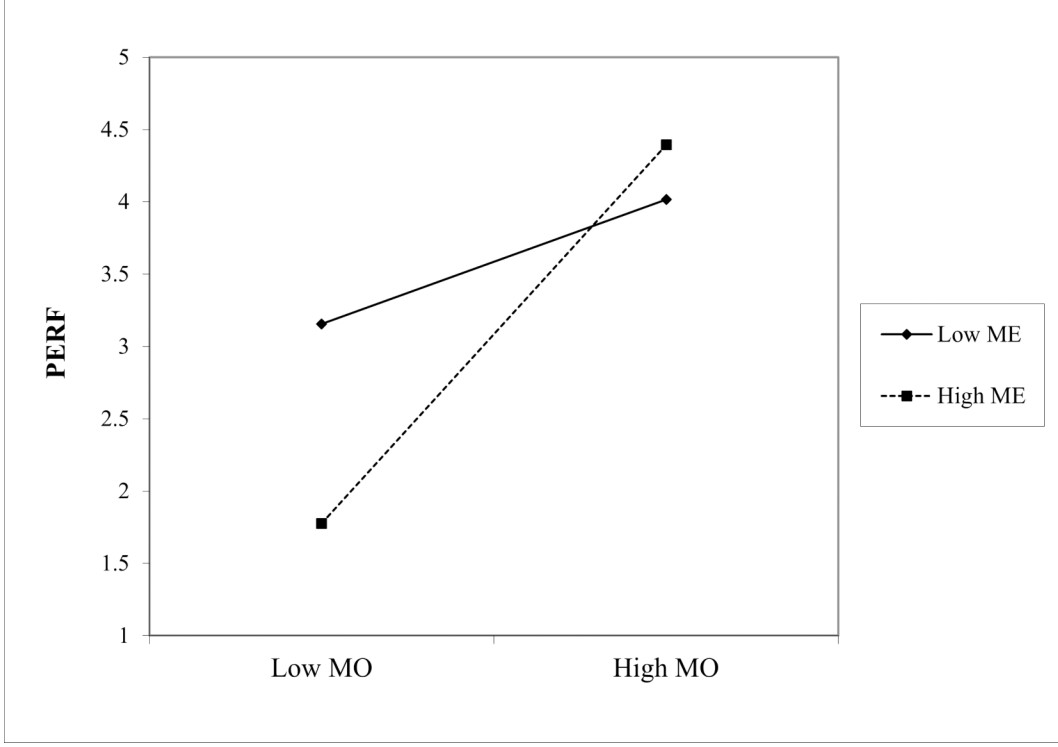

**Fig 4. Moderating effect of ME between MO and PERF.**

of SMC: Marketing Culture, Marketing Learning, and Marketing Operation, thus supporting Hypothesis Ha and all six sub-hypotheses (Ha1–Ha6).

This finding aligns with Vorhies et al. [36], who conceptualized marketing capabilities as a form of higher-order dynamic capability and extended the theoretical model of "exploration-driven capability reconstruction" to the context of culturally embedded traditional enterprises. Consistent with Zhang and Qiu [63], who emphasized the complementary roles of exploratory and exploitative innovation, our results demonstrate how exploration enhances learning systems, whereas exploitation strengthens process efficiency.

Lyu et al. [66] linked ambidextrous leadership to corporate sustainability; this study broadens that view by illustrating how ambidextrous marketing fosters cultural consensus and knowledge absorption, enabling embedded, synergistic development. This echoes Zhang et al.'s [21] observation that firms achieve "process stability through exploitation and innovation via exploration" under big data capability.

Furthermore, our study supports Zhang et al. [19], who argued that ambidextrous strategies promote long-term competitive advantage by showing that Time-Honored Brands naturally integrate exploration and exploitation within deeply rooted brand cultures. Importantly, no strong trade-offs or structural tensions were observed between the two strategies. Instead, the evidence points to a highly integrated, nested relationship—what we term a "culturally buffered ambidextrous synergy."

Notably, this synergy manifests as a dual pathway of "preserving core foundations while pioneering innovation," revealing how traditional Chinese firms harmonize between heritage and modernity. These findings offer novel empirical support for the coexistence and integration of dual capabilities in a non-adversarial, culturally embedded context.

**5.2.2. The impact of sustainable marketing capability on corporate performance.** The findings reveal that Sustainable Marketing Capability (SMC) has a significant positive effect on Corporate Performance, thereby supporting

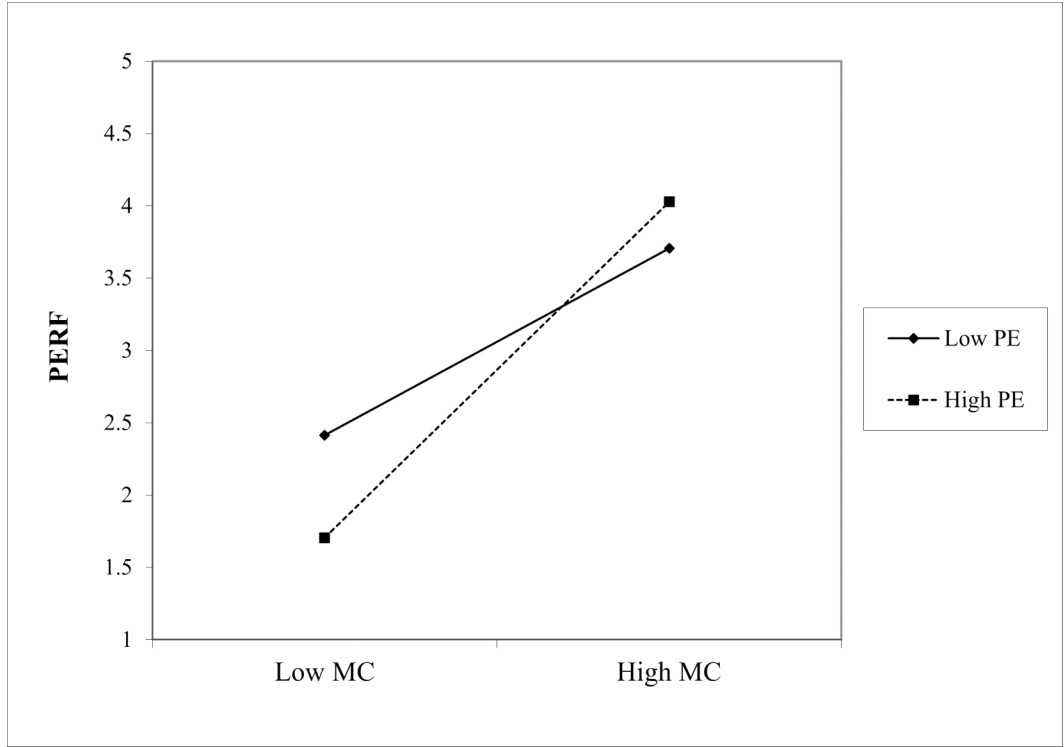

**Fig 5. Moderating effect of PE between MC and PERF.**

Hypothesis Hb and its sub-hypotheses (Hb1–Hb3). Among the three dimensions of SMC, Marketing Culture fosters internal goal alignment and enhances customer satisfaction; Marketing Learning facilitates knowledge integration and cross-functional innovation; and Marketing Operation improves market responsiveness and customer value through standardized and efficient execution. These results are consistent with Xu and Wang [24], who emphasized the multifaceted contributions of SMC to performance.

The observed relationships are further supported by existing literature. For instance, Mumel et al. [71] highlighted that market orientation plays a pivotal role in enhancing customer loyalty and securing competitive advantage. Sampaio et al. [72] argued that a dynamic balance between exploration and exploitation is critical for long-term performance sustainability. Kunieda and Takashima [26] demonstrated that service quality mediates the relationship between market orientation and firm performance. Together, these studies strengthen the theoretical foundation for linking SMC dimensions to firm-level outcomes and validate the performance-enhancing function of the SMC.

Situated within the specific context of China's time-honored enterprises, the results suggest that even legacy firms—often constrained by limited resources, organizational inertia, and slower innovation cycles—can achieve stable and meaningful performance improvements through effective use of SMC. By leveraging cultural alignment, accumulated knowledge, and operational efficiency, the SMC provides a robust internal mechanism for sustainable value creation. Notably, Marketing Culture is critical in ensuring consistent brand delivery and performance reliability amid increasing market volatility.

More importantly, the simultaneous support for both Ha and Hb reinforces the core proposition advanced by Boumgarden et al. [17] and Lee & Kim [18] that long-term success depends on an organization's ability to simultaneously engage in exploration and exploitation. The empirical results of this study demonstrate that even culturally embedded,

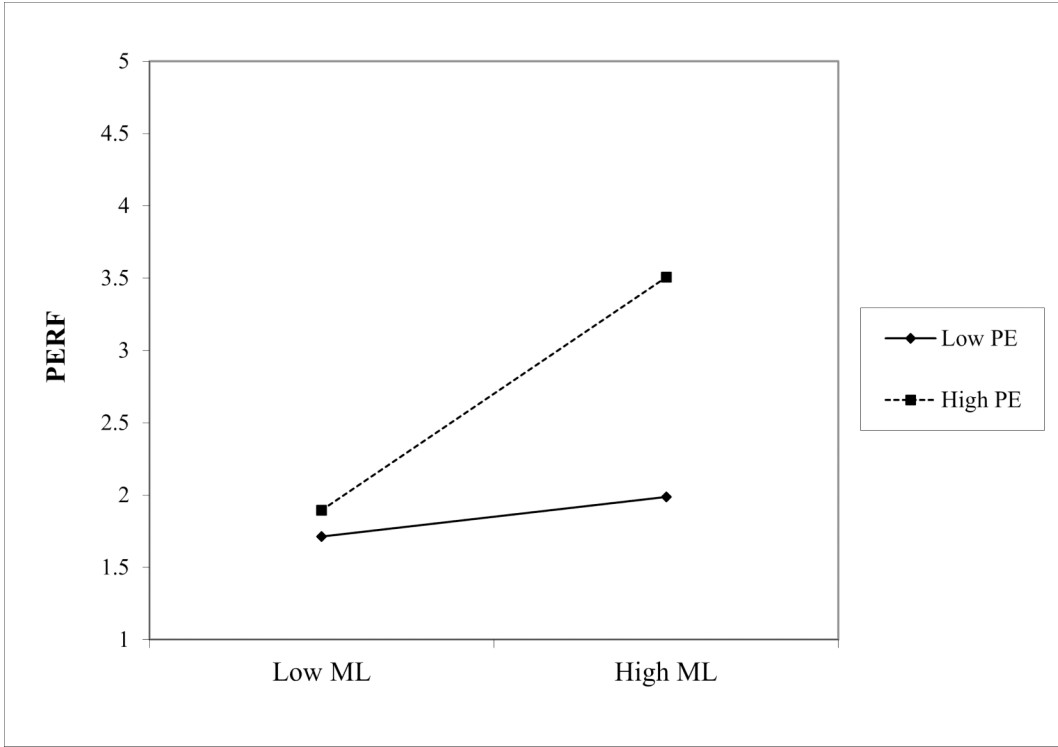

**Fig 6. Moderating effect of PE between ML and PERF.**

resource-constrained heritage brands can achieve breakthrough performance by adopting a dual-path strategy and a capability-based development model.

These findings extend the applicability of the "Ambidexterity → Capability → Performance" framework, demonstrating its relevance in traditional and locally grounded organizational settings.

**5.2.3. The mediating role of sustainable marketing capability.** This study further confirms the mediating role of Sustainable Marketing Capability (SMC) in the relationship between Marketing Exploration (PLOR), Marketing Exploitation (PLOI), and Corporate Performance (PERF), thereby validating hypothesis Hc. Among the three dimensions of SMC, Marketing Learning exhibited the strongest mediating effect, suggesting that knowledge absorption, experiential continuity, and cross-functional collaboration are key mechanisms through which marketing strategies are translated into performance outcomes.

These findings are consistent with He et al. [45], who identified marketing capabilities as critical mediators between market-oriented innovation and firm performance. The results also extend the ambidextrous innovation mediation model proposed by Zhang et al. [21], demonstrating its applicability in the culturally embedded context of Time-Honored Brands. Similarly, Lyu et al. [66] introduced a sequential mediation model involving green creativity and green product innovation, further highlighting the strategic relevance of capability-based mediation in driving sustainable development.

Notably, the mediating effects varied across the three SMC dimensions. Marketing Culture plays a more prominent role in the PLOR-to-PERF pathway, reflecting how cultural alignment and shared values facilitate the transformation of exploratory strategies into competitive advantage. In contrast, Marketing Operation exerts greater influence on the PLOI-to-PERF pathway, emphasizing the importance of process optimization and execution in leveraging exploitative strategies.

This structural divergence aligns with the differentiated mediation framework proposed by Asree [73], suggesting that the impact of ambidextrous marketing is not uniform but channeled through distinct capability dimensions. These findings

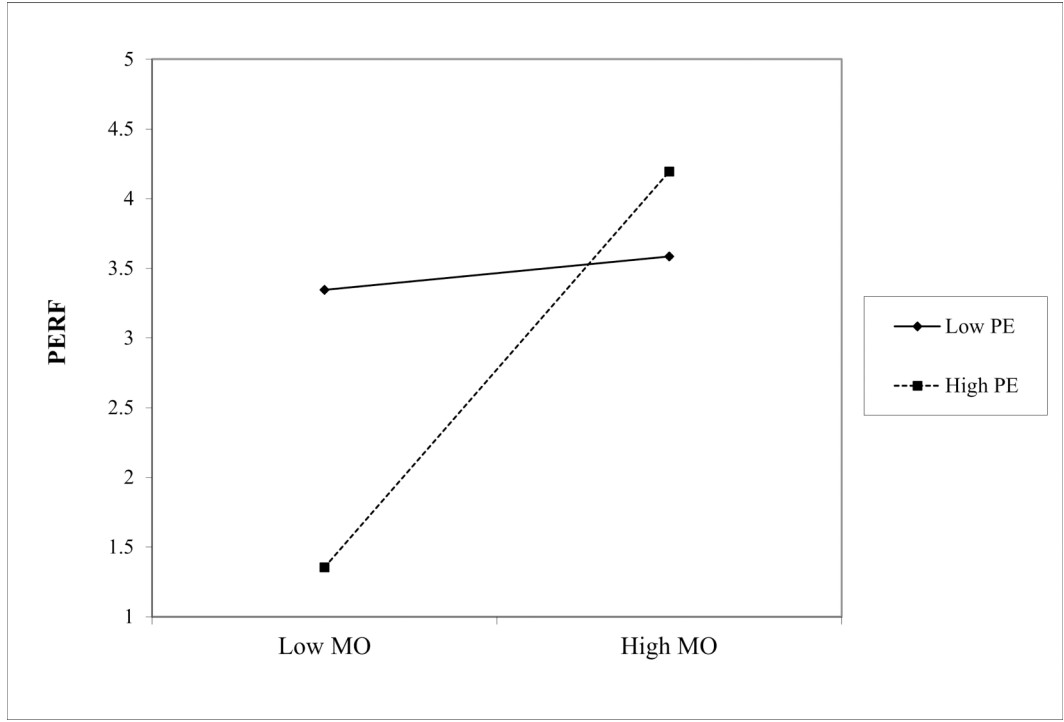

**Fig 7. Moderating effect of PE between MO and PERF.**

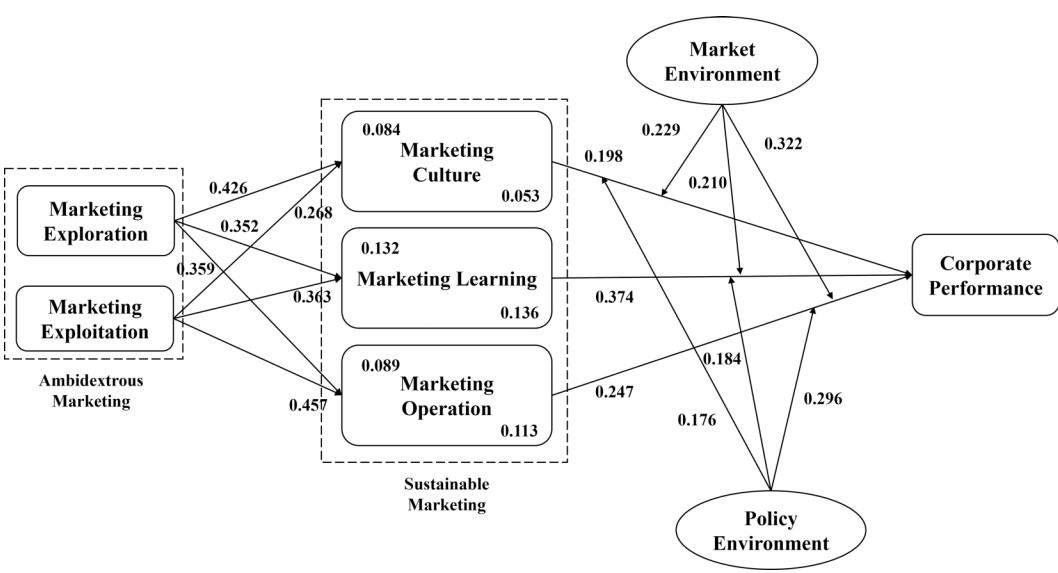

**Fig 8. Research results.**

underscore the nuanced pathways through which exploration and exploitation strategies contribute to performance, further validating the "strategy → capability → performance" logic embedded in this study's theoretical model.

**5.2.4. Moderating effects of market and policy environments.** Using a latent interaction modeling approach, this study tested the moderating effects of the Market Environment and Policy Environment on the pathway from Sustainable Marketing Capability to Corporate Performance, thereby confirming Hypotheses Hd and He. Both environmental factors were found to be significant positive moderators, although their influence varied across the three dimensions of SMC.

Specifically, under conditions of heightened competition and shifting consumer demand, the performance conversion effect of Marketing Operation was most pronounced. This finding indicates that a well-executed and adaptive operational system is a critical asset for time-honored firms navigating external uncertainty, consistent with Peng et al. [77]. In addition, the enhanced contribution of Marketing Learning under dynamic market conditions supports the argument by Appiah-Nimo and Chovancová [57], who demonstrated that market orientation strengthens the strategy–performance linkage.

From a policy perspective, the Policy Environment exhibited particularly strong moderating effects on marketing culture and learning. This validates Luan et al.'s [78] insights on policy stability and capability embedding, and echoes Tshuma's [79] conclusion that "green policy" frameworks can activate internal adaptation mechanisms. These results suggest that institutional environments are not merely external constraints or enablers; they also play a pivotal role in internalizing cultural values and learning capabilities within traditional enterprises.

Taken together, Market Environment and Policy Environment function as external "filtering mechanisms" that shape the effectiveness of the transformation of internal capabilities into tangible performance outcomes. These findings contribute to an enriched understanding of the interaction between organizational capabilities and environmental contexts, offering empirical support for the integration of Institutional Theory with dynamic capability frameworks.

From a managerial perspective, this suggests that strategic breakthroughs for culturally embedded enterprises depend not only on internal capability building but also on the firm's ability to align with—and adapt to—market and policy shifts. This dual alignment enables traditional firms to transform their inherited advantages into sustainable performance in volatile environments.

## 5.3. Managerial implications

This study proposes an empirically validated framework that links Ambidextrous Marketing, Sustainable Marketing Capability (SMC), and Corporate Performance. The findings offer actionable guidance for brand managers, strategic decision-makers, and policymakers involved in revitalizing and modernizing China's Time-Honored Brands.

Modern business operations begin and end with marketing. Therefore, marketing management should be regarded as a central strategic function in enterprise governance. From the perspectives of contemporary marketing theory and systems thinking, firms aiming for long-term success must cultivate an internal, organization-wide driver of competitiveness: the SMC. The closed-loop model of Ambidextrous Marketing → SMC → Performance underscores the critical role of SMC in supporting strategic transformation, particularly within traditional enterprises.

For business managers, a key challenge is balancing exploration and exploitation amid resource constraints. This study demonstrates that these two strategic logics can coexist through a combination of experience-based learning and opportunity recognition. Managers are encouraged to implement "lightweight exploration" approaches, such as using digital platforms, launching brand extensions, or experimenting with product prototypes, to engage new market opportunities without disrupting core operations. This allows firms to remain agile while ensuring operational continuity.

For strategic decision-makers, the three dimensions of SMC—Marketing Culture, Marketing Learning, and Marketing Operation— uniquely contribute to performance. Sustained competitiveness depends on the long-term accumulation of capabilities, particularly by building organizational learning systems and reinforcing cultural identity. Given that Marketing Learning emerged as a key mediating factor, firms should establish cross-functional communication channels and institutionalize internal processes that facilitate experience sharing, knowledge absorption, and process renewal. At the

same time, revitalizing brand narratives that integrate traditional cultural values with modern consumer preferences can strengthen emotional engagement and translate intangible heritage into measurable value.

For policymakers, this study highlights the significant moderating effects of both market and policy environments on the capability–performance relationship. This suggests that external incentives play a crucial role in activating legacy firms' internal potential. Policy frameworks should avoid standardized resource allocation or rigid assessment criteria. Instead, support mechanisms should be tailored to reflect specific industry characteristics, brand maturity levels, and regional development contexts. The recommended initiatives include subsidies for digital transformation, tax incentives for cultural heritage preservation, and the creation of regional platforms for brand incubation and innovation.

From a capability-building perspective, Time-Honored Brands should adhere to two core principles: rhythmic coordination and cultural embeddedness. Rhythmic coordination refers to the dynamic alignment of speed, implementation, and resource allocation between exploratory and exploitative activities. Cultural embeddedness ensures that all capability development efforts are grounded in a brand's historical and cultural identity. This approach enables firms to pursue sustainable growth through a "core-preserving, innovation-driven" transformation path that respects their legacy, while embracing change.

## 5.4. Theoretical contributions

This study addresses the challenge of capability development in China's Time-Honored Brands by constructing and empirically validating a multi-path, multi-mechanism theoretical framework, grounded in the overarching logic of "corporate adaptability → sustainable capability → corporate performance." This study offers four main theoretical contributions.

First, this study develops a closed-loop model linking Ambidextrous Marketing → Sustainable Marketing Capability (SMC) → Corporate Performance. While previous research has often focused on the direct impact of ambidextrous marketing on outcomes, this study highlights the mediating role of capability development. By integrating Marketing Exploration and Exploitation through the three-dimensional structure of SMC, it uncovers a meso-level pathway of "capability synergy → structural reconstruction → performance transformation." The findings demonstrate that ambidextrous marketing is not confined to high-tech or emerging sectors; it is equally applicable to culturally embedded legacy firms, offering strong explanatory power in traditional business contexts.

Second, this study extends the structural and functional boundaries of SMC. By decomposing SMC into Marketing Culture, Marketing Learning, and Marketing Operation, it clarifies the internal dynamics of capability formation and reveals the differentiated performance contributions of each sub-dimension. The proposed framework of "cultural empowerment → learning transformation → operational execution" enhances the theoretical utility of SMC as a mediating variable. Moreover, the study reinforces that the effect of ambidextrous marketing on performance is indirect and capability-driven, offering a deeper understanding of how strategic actions translate into organizational outcomes.

Third, by incorporating the moderating roles of market and policy environments, this study enriches the explanation of context-sensitive mechanisms in capability-performance conversion. The results support the "environment → capability → performance" logic and show that the effectiveness of internal capabilities is contingent on the level of market dynamism and institutional support. These insights enhance the applicability of institutional and contingency theories in capability-based research, particularly in traditional and regulated industries.

Finally, this study fills a significant empirical gap by focusing on Time-Honored Brands in Southwest China, a region underrepresented in quantitative capability research. While the existing literature on heritage brands has largely centered on qualitative issues such as brand perception or cultural storytelling, this study is among the first to empirically test a capability development model in this context. This confirms that performance gains in traditional enterprises can be achieved through a combined mechanism of capability mediation and contextual moderation. These findings contribute novel theoretical insights to the literature on heritage brand management and provide a foundation for future research on strategy optimization and governance reform in culturally embedded firms.

### 5.5. Limitations and future research directions

This study had several limitations that present opportunities for future research. First, the cross-sectional research design limits the ability to capture the dynamic evolution of marketing capabilities over time. While the current model reveals the structural relationships among Ambidextrous Marketing, Sustainable Marketing Capability, and Corporate Performance, it does not address how these relationships develop or shift across different stages of organizational growth. Future research could adopt longitudinal designs or rolling panel surveys to explore temporal dynamics, including feedback loops, adaptation processes, and time-lagged effects within the capability–performance relationship.

Second, this study emphasizes structural and internal variables, with limited attention to soft factors such as organizational culture, consumer emotion, and brand perception. However, as culturally embedded entities, Time-Honored Brands are shaped not only by internal systems but also by consumer identity, emotional attachment, and mechanisms of cultural transmission. Future research could incorporate constructs such as brand nostalgia, cultural congruence, and consumer-brand resonance to build a cross-actor capability framework that connects internal capability building with external value co-creation.

Such an expanded approach would bridge internal marketing processes with external brand engagement, offering a more holistic understanding of how heritage firms sustain their competitiveness in emotionally and culturally complex market environments.

## 6. Conclusion

This study investigates how firms can develop Sustainable Marketing Capability (SMC) to improve corporate performance, focusing on three core research questions. The empirical results offer clear and comprehensive answers to these questions.

First, in response to RQ1, the study constructs and empirically validates a structural model linking Marketing Exploration and Exploitation→Marketing Culture, Marketing Learning, and Marketing Operation→Corporate Performance. This framework clarifies the antecedents and outcomes of SMC. The findings reveal that ambidextrous marketing practices effectively stimulate cultural alignment, knowledge transformation, and operational optimization, thereby fostering a sustainable competitive advantage. The tripartite structure of SMC—representing cognitive, learning, and executional capabilities—functions both independently and synergistically, forming an endogenous capability chain from internal identity to market impact.

Second, addressing RQ2, the study applies a bootstrap mediation analysis to confirm the significant mediating effects of all three SMC dimensions in the pathways from Marketing Exploration and Exploitation to performance. These results demonstrate that SMC is not an outcome in itself, but a strategic mediator that bridges marketing input and performance realization. This meso-level mechanism highlights how strategy formulation is transformed into organizational outcomes, extending the applicability of ambidexterity theory to legacy firms and directly addressing theoretical concerns regarding unpacking the underlying capability logic.

Third, in response to RQ3, the study employs latent moderated structural equation modeling to verify that both the Market Environment and Policy Environment positively moderate the relationship between SMC and performance. The differential strength of moderation across the SMC dimensions suggests that firms must consider environmental dynamism and institutional alignment when designing internal capability systems. These findings validate the contextual embeddedness of the capability–performance relationship and support the integration of capability and institutional theories.

In summary, this study provides a robust theoretical and empirical foundation for understanding how time-honored enterprises can navigate dynamic environments and drive performance through ambidextrous marketing and sustainable capability building.

# Supporting information

**S1 File. Original data.**

(SAV)

## Author contributions

**Conceptualization:** Xiangyu Li.

**Data curation:** Xiangyu Li.

**Formal analysis:** Danaikrit Inthurit.

**Investigation:** Xiangyu Li, Linjun Qiu.

**Methodology:** Xiangyu Li, Maochun Wu.

**Supervision:** Danaikrit Inthurit.

**Validation:** Danaikrit Inthurit.

**Writing – original draft:** Xiangyu Li, Linjun Qiu.

**Writing – review & editing:** Xiangyu Li, Maochun Wu.

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
