## [Decision Letter · Decision Letter 0]

PONE-D-24-48512Breaking the deadlock: a study on the pathway and effects of reshaping the sustainable marketing capability of Chinese time-honored brandsPLOS ONE

Dear Dr. Li,

Thank you for submitting your manuscript to PLOS ONE. After careful consideration, we feel that it has merit but does not fully meet PLOS ONE’s publication criteria as it currently stands. Therefore, we invite you to submit a revised version of the manuscript that addresses the points raised during the review process. Please submit your revised manuscript by May 26 2025 11:59PM. If you will need more time than this to complete your revisions, please reply to this message or contact the journal office at plosone@plos.org . Please include the following items when submitting your revised manuscript:

We look forward to receiving your revised manuscript.

Kind regards,

Manuel Herrador, Ph.D.

Academic Editor

PLOS ONE

Journal Requirements:

3. We note that your Data Availability Statement is currently as follows: 

“All relevant data are within the manuscript and its Supporting Information files”

5. Please ensure that you refer to Figure 1 in your text as, if accepted, production will need this reference to link the reader to the figure.

**Additional Editor Comments:**

Dear Authors,

Given the reviewers' feedback, I recommend Major Revisions to address the critical concerns before acceptance.

Best regards

Reviewers' comments:

Reviewer's Responses to Questions

**Comments to the Author**

1. Is the manuscript technically sound, and do the data support the conclusions?

Reviewer #1: Yes

Reviewer #2: Partly

2. Has the statistical analysis been performed appropriately and rigorously? 

Reviewer #1: Yes

Reviewer #2: No

3. Have the authors made all data underlying the findings in their manuscript fully available?

Reviewer #1: Yes

Reviewer #2: No

4. Is the manuscript presented in an intelligible fashion and written in standard English?

Reviewer #1: Yes

Reviewer #2: No

5. Review Comments to the Author

Reviewer #1: The article 'Breaking the deadlock: a study on the pathway and effects of reshaping the sustainable marketing capability of Chinese time-honored brands' presents a highly engaging and valuable contribution to the field. It was a pleasure to review this work.

The authors address a timely and significant topic, focusing on the sustainability of traditional brands in a rapidly evolving business environment. The theoretical model developed in this study is well-grounded, effectively integrating concepts of ambidextrous marketing, sustainable marketing capabilities, and corporate performance. This integration provides a comprehensive framework for understanding the challenges and opportunities faced by time-honored brands in China.

Methodologically, the research is robust, employing advanced statistical techniques, including structural equation modeling. This approach lends credibility to the findings and enhances the overall rigor of the study. Furthermore, the insights derived from this research offer valuable guidance for managers of traditional brands, potentially aiding in the development of more effective and sustainable marketing strategies.

While the study is strong in many aspects, there is room for improvement. Specifically, the paper would benefit from a more in-depth discussion of the particularities of the sample of brands studied. A more detailed exploration of the unique characteristics of these time-honored brands and their specific market contexts would enhance the generalizability and applicability of the findings.

Overall, this article makes a significant contribution to our understanding of sustainable marketing capabilities in the context of traditional Chinese brands. It not only advances theoretical knowledge but also provides practical implications for brand management in an increasingly complex business landscape.

Reviewer #2: The subject of the article is interesting but major improvements are needed for making this article suitable to publication. Please find below my remarks:

1. The authors do not explain enough the concepts used in the paper like: Ambidextrous marketing, PLOR, PLOI, MC, ML, MO. The abbreviations are excesively used so that the readers beacome really confused.

2. The methodology is not enough explained. How the questionnaire looks like? The authors declare that the final sample includes 96 enterprises, but the data were collected from “352 valid questionnaires were retained from 47 companies”. Why the authors present the sample characterstics in the Results section?

3. Results. It is not very clear how the dimensions presented in Table 3 have been obtained? Were they designed from the beginning or the EFA was used?

4. In the Conclusion section the research questions are not addressed any more. The contribution of the research should be emphasized. The implications of the research results for theory and practice are not clearly described in the article. A detailed explanations of the author’s recommendations should be included.

In my opinion, the authors have to make efforts to clarify many aspects of the article and report them into a coherent manner.

6. PLOS authors have the option to publish the peer review history of their article (what does this mean? ). If published, this will include your full peer review and any attached files.

**Do you want your identity to be public for this peer review?** For information about this choice, including consent withdrawal, please see our Privacy Policy .

Reviewer #1: **Yes: ** Patricia Regina Caldeira Daré Artoni

Reviewer #2: No

---

## [Author Response · Author response to Decision Letter 1]

26 May 2025

Response to Editorial Requirements

We thank the Academic Editor and the journal office for the opportunity to revise and resubmit our manuscript entitled "Breaking the deadlock: a study on the pathway and effects of reshaping the sustainable marketing capability of Chinese time-honored brands" (Manuscript ID: PONE-D-24-48512).

In accordance with the submission guidelines, we have completed the following:

1. We have uploaded a revised manuscript with tracked changes to highlight all modifications made, as well as a clean version of the manuscript without tracked changes for the final evaluation.

2. We have provided all relevant data required to replicate the results of our study, including the raw values and summary statistics. Since our study is based on a quantitative survey design and does not involve laboratory-based experimental procedures, it is not applicable to upload experimental protocols to protocols.io. However, we have ensured the reproducibility of our results by fully sharing the dataset and measurement instruments used in our analysis.

3. No changes were made to the financial disclosure statement. The original disclosure remains accurate and up-to-date.

4. To enhance transparency and facilitate editorial review, we have color-coded all revisions in the manuscript according to their source. Specifically, responses to the Academic Editor and journal requirements are marked in blue font, revisions addressing Reviewer #1’s comments are marked in purple font, and those responding to Reviewer #2 are marked in red font. In addition, changes made by the authors upon internal post-revision review—such as clarifications or minor corrections not directly prompted by reviewers—are marked in dark red font. These changes can be found in the file titled “Revised Manuscript with Track Changes”.

Below, we provide a point-by-point response to the editor’s and each of the reviewers’ comments, with the corresponding revisions clearly marked in the revised manuscript.

Response to Journal Requirement

Editor Comment #1:

Response:

We thank you for this reminder. We have carefully reviewed the PLOS ONE formatting guidelines and revised the manuscript accordingly to ensure compliance. Specifically, we adjusted the structure, section headings, figure referencing, reference format, and English grammar to align with the PLOS ONE style templates. We have also renamed our files according to the journal’s file-naming conventions.

Editor Comment #2:

Please provide additional details regarding the participant consent. In the ethics statement in the Methods and online submission information, please ensure that you have specified the following:

(1) whether consent was informed and

(2) what type you obtained (for instance, written or verbal, and if verbal, how it was documented and witnessed). If your study included minors, state whether you obtained consent from parents or guardians.

Response:

We appreciate this important comment. We have now added a complete ethics statement to the Methodology section of the manuscript (see 3.2 Ethics Statement, which is located from line 447 to line 453, on page 18-19 of the manuscript). Specifically, we confirmed that informed verbal consent was obtained from all participants prior to survey administration. Participants were clearly informed about the study’s objectives, procedures, and voluntary nature, and they provided verbal consent before proceeding. No minors were involved in this study. The consent process was reviewed and approved by the Ethics Committee of Chuxiong Normal University.

Additionally, we confirm that this manuscript does not involve any retrospective studies of medical records or archived biological samples.

Editor Comment #3:

Please confirm at this time whether or not your submission contains all raw data required to replicate the results of your study. Authors must share the “minimal data set” for their submission. If your submission does not contain these data, please either upload them as Supporting Information files or deposit them to a stable, public repository and provide us with the relevant URLs, DOIs, or accession numbers.

Response:

We thank you for highlighting this important requirement. We confirm that we have provided all the data necessary to replicate the findings of our study. This includes the raw scores underlying the descriptive statistics, inferential tests, and structural equation modeling. These data have been uploaded as Supporting Information: [original data.xlsx and original data.sav]. The data were fully anonymized prior to sharing, and no personally identifiable information was included.

Editor Comment #4:

When completing the data availability statement of the submission form, you indicated that you will make your data available on acceptance. We strongly recommend all authors decide on a data sharing plan before acceptance. Please note that your entire data will need to be made freely accessible if your manuscript is accepted for publication.

Response:

We fully support ’s open data policy of PLOS ONE. We confirm that we have now prepared a complete data-sharing plan and agree to make all relevant data freely accessible upon publication. The minimal dataset necessary to replicate the findings of this study has been compiled and uploaded as Supporting Information. These files include raw and summary data and will be available without restrictions upon publication.

Editor Comment #5:

Please ensure that you refer to Figure 1 in your text, as, if accepted, production will need this reference to link the reader to the figure.

Response:

Thank you for this helpful reminder. We have now added explicit in-text references to Figure 1 in the manuscript. The reference appears on page 17, line 429.

Response to Reviewer

Reviewer #1 Comment:

The article 'Breaking the deadlock: a study on the pathway and effects of reshaping the sustainable marketing capability of Chinese time-honored brands' presents a highly engaging and valuable contribution to the field. It was a pleasure to review this work.

The authors address a timely and significant topic, focusing on the sustainability of traditional brands in a rapidly evolving business environment. The theoretical model developed in this study is well-grounded, effectively integrating concepts of ambidextrous marketing, sustainable marketing capabilities, and corporate performance. This integration provides a comprehensive framework for understanding the challenges and opportunities faced by time-honored brands in China.

Methodologically, the research is robust, employing advanced statistical techniques, including structural equation modeling. This approach lends credibility to the findings and enhances the overall rigor of the study. Furthermore, the insights derived from this research offer valuable guidance for managers of traditional brands, potentially aiding in the development of more effective and sustainable marketing strategies.

While the study is strong in many aspects, there is room for improvement. Specifically, the paper would benefit from a more in-depth discussion of the particularities of the sample of brands studied. A more detailed exploration of the unique characteristics of these time-honored brands and their specific market contexts would enhance the generalizability and applicability of the findings.

Response:

We sincerely thank Reviewer 1 for the thoughtful and encouraging feedback on the theoretical design, methodological rigor, and practical implications of our study.

In response to your valuable suggestions regarding the discussion of sample characteristics and contextual uniqueness, we have made the following revisions:

In the Introduction section, we added a new paragraph following the “Research Contributions” to explicitly highlight the regional and cultural embeddedness of the sample firms. This addition clarifies how the specific historical, structural, and market conditions of Southwest China shape the developmental challenges and opportunities faced by China Time-Honored Brands (see page 5, lines 120–126).

In the newly added Discussion section (5.1 Sample Characteristics and Contextual Influence), we elaborate on the industry distribution, cultural attributes, innovation dynamics, and policy environment of the surveyed firms. We show how these factors interact with marketing exploration and exploitation capabilities and how they influence the sustainable development pathways for legacy brands (see page 33-34, lines 706–732).

These revisions aim to strengthen the contextual richness of the study and improve the generalizability and applicability of the findings, as you have insightfully suggested. We greatly appreciate your constructive feedback.

Response to Reviewer

Reviewer #2 Comment 1

The authors do not explain enough the concepts used in the paper like: Ambidextrous marketing, PLOR, PLOI, MC, ML, MO. The abbreviations are excesively used so that the readers beacome really confused.

Response:

We sincerely thank the reviewer for highlighting this important aspect. We fully understand that the overuse of abbreviations without adequate conceptual explanation can hinder readability, especially for readers who may not be familiar with these terms.

1. In response, we carefully and comprehensively reviewed the manuscript to ensure that all key concepts are clearly defined upon their first appearance. Specifically, we have provided explicit definitions for the following terms:

Ambidextrous Marketing (line 175-178; page 7), Marketing Exploration (line 178-180; page 7), Marketing Exploitation (line 180-182; page 7), Sustainable Marketing Capability (line 238-242; page 10), Marketing Culture (line 247-248; page 10), Marketing Learning (line 249-250; page 10), Marketing Operation (line 251-252; page 10), Corporate Performance (line 307-318; page12-13), Market Environment (line 382-384; page 15), and Policy Environment (line 385-388; page 15-16).

2. Regarding the use of abbreviations, we have revised the manuscript to follow a consistent and reader-friendly strategy. Specifically, abbreviations are applied primarily in sections where a term appears frequently and repetitively to improve clarity without compromising academic rigor. Our approach follows three main principles.

In each major section (e.g., Introduction, Literature Review, Methodology), we spell out the full term with its abbreviation in parentheses at first mention. Thereafter, only the abbreviation is used in areas where the term occurs repeatedly.

When discussing the theoretical foundations or measurement design, we consistently use the full term again to preserve academic precision and support reader comprehension.

In all tables, abbreviations are retained to enhance layout clarity and avoid space constraints, particularly in wide-format ones.

To further facilitate transparency in the revision process, all locations where abbreviations were modified—whether full terms were added, abbreviations replaced, or adjustments were made for consistency—are clearly marked in red font in the revised manuscript.

We hope that this approach resolves your concerns and improves the manuscript’s readability. If further refinements are needed, we would appreciate your suggestions and would be happy to revise accordingly.

Reviewer #2 – Comment 2

2. The methodology is not enough explained. How the questionnaire looks like? The authors declare that the final sample includes 96 enterprises, but the data were collected from “352 valid questionnaires were retained from 47 companies”. Why the authors present the sample characterstics in the Results section?

Response:

Thank you very much for your detailed and constructive comments on our methodology and sampling. We sincerely appreciate the opportunity to clarify the following issues.

(1) Regarding the methodology, we have added a new subsection (Section 3.1 Research Design, line 436-445; page 18) at the beginning of the Methodology section. This section now clearly explains our overall research strategy: a quantitative approach using structural equation modeling as the primary analysis technique, supported by bootstrapping for mediation testing and latent moderated structural equations for moderation analysis. This aligns directly with our research questions.

(2) Regarding the questionnaire structure, we would like to clarify that the measurement items corresponding to each latent variable are already displayed in Table 3 under the “Variables and Items” column, which fully reflects the content of the questionnaire. The full version of the questionnaire was submitted as a supplementary material during the initial submission. If needed, we are happy to upload the questionnaire again, along with this revised manuscript, for your convenience.

(3) Regarding sample inconsistency, we apologize for the confusion caused by the earlier wording. The figure of 96 enterprises refers to the planned sample size, calculated based on Cochran’s formula and proportionally allocated across the provinces. However, due to constraints such as enterprise willingness and accessibility, valid responses were ultimately collected from 47 enterprises, resulting in 352 valid response questionnaires. We have revised Sections 3.4 Sampling (line 504-507; page 21) and 4.1 Sample Characteristics (line 524-530; page 22) to clearly distinguish between the planned and actual samples to avoid further ambiguity.

(4) Regarding the placement of the sample characteristics section, we agree that clarity of structure is important. The reason we presented Sample Characteristics under the Results section (Section 4.1 Sample Characteristics) is that these reflect the actual structure of the valid responses received, which could not be known in advance during the planning stage. In contrast, the sample information in the Methodology section (Section 3.4 Sampling) reflects the intended framework. We believe that this division is consistent with empirical research practice; however, if the reviewer suggests consolidation, we will be happy to make further adjustments.

Once again, we appreciate your thoughtful feedback, which has helped us improve the clarity and precision of the manuscript.

Reviewer #2 – Comment 3

Results. It is not very clear how the dimensions presented in Table 3 have been obtained? Were they designed from the beginning or the EFA was used?

Response:

Thank you for your question regarding the origin of the dimensions in Table 3. We would like to clarify that the measurement items and dimensions used in this study were derived from previously validated and widely accepted scales in the literature. Where necessary, we made minor adjustments to better reflect the research context of time-honored brands in southwestern China. This theoretical and literature-based approach was chosen to ensure content validity and comparability with previous studies.

To ensure reliability and validity, we conducted expert reviews and a pilot study before formal data collection. Subsequently, we applied confirmatory factor analysis (CFA) to verify the factor structure and assess the measurement quality of the model.

Therefore, we did not perform exploratory factor analysis (EFA) in this study. As this is a theory-driven research design, the measurement dimensions were conceptually and theoretically defined from the outset rather than derived from empirical data. This rationale is clearly explained in Section 3.3 Measurement (lines 456–462; page 19) of the revised manuscript. Additionally, to enhance clarity and transparency, we have carefully marked the source references for each measurement scale in red font within the 3.3 Measurement. We hope that this approach meets your expectations, and we would be grateful for any further suggestions you may have.

Reviewer #2 – Comment 4

Reviewer Comment:

In the Conclusion section the research

---

## [Editor Report · Decision Letter 1]

Breaking the deadlock: a study on the pathway and effects of reshaping the sustainable marketing capability of Chinese time-honored brands

PONE-D-24-48512R1

Dear Dr. Li,

We’re pleased to inform you that your manuscript has been judged scientifically suitable for publication and will be formally accepted for publication once it meets all outstanding technical requirements.

Kind regards,

Manuel Herrador, Ph.D.

Academic Editor

PLOS ONE

Additional Editor Comments (optional):

Dear authors,

I am pleased to inform you that both reviewers are satisfied with the paper in its current form.

Well done, and thank you for your contribution to PLOS One.

Best regards
---

## [Editor Report · Acceptance letter]

PONE-D-24-48512R1

PLOS ONE

Dear Dr. Li,

I'm pleased to inform you that your manuscript has been deemed suitable for publication in PLOS ONE. Congratulations! Your manuscript is now being handed over to our production team.

Kind regards,

on behalf of

Dr. Manuel Herrador

Academic Editor

PLOS ONE